# Structural basis of tethered agonism of the adhesion GPCRs ADGRD1 and ADGRF1

Xiangli Qu[1,2,8], Na Qiu[1,2,8], Mu Wang[1,3,8], Bingjie Zhang[4,8], Juan Du[5], Zhiwei Zhong[6], Wei Xu[1,2], Xiaojing Chu[1], Limin Ma[1], Cuiying Yi[1], Shuo Han[1,2], Wenqing Shui[4✉], Qiang Zhao[1,2,7✉] & Beili Wu[1,2,3,5,6✉]

Adhesion G protein-coupled receptors (aGPCRs) are essential for a variety of physiological processes such as immune responses, organ development, cellular communication, proliferation and homeostasis[1–7]. An intrinsic manner of activation that involves a tethered agonist in the N-terminal region of the receptor has been proposed for the aGPCRs[8,9], but its molecular mechanism remains elusive. Here we report the G protein-bound structures of ADGRD1 and ADGRF1, which exhibit many unique features with regard to the tethered agonism. The stalk region that proceeds the first transmembrane helix acts as the tethered agonist by forming extensive interactions with the transmembrane domain; these interactions are mostly conserved in ADGRD1 and ADGRF1, suggesting that a common stalk–transmembrane domain interaction pattern is shared by members of the aGPCR family. A similar stalk binding mode is observed in the structure of autoproteolysis-deficient ADGRF1, supporting a cleavage-independent manner of receptor activation. The stalk-induced activation is facilitated by a cascade of inter-helix interaction cores that are conserved in positions but show sequence variability in these two aGPCRs. Furthermore, the intracellular region of ADGRF1 contains a specific lipid-binding site, which proves to be functionally important and may serve as the recognition site for the previously discovered endogenous ADGRF1 ligand synaptamide. These findings highlight the diversity and complexity of the signal transduction mechanisms of the aGPCRs.

The aGPCR family (class B2) is by far the least understood class of GPCRs; most of its members are still orphan receptors and are not yet pharmacologically targeted. These receptors have a unique molecular structure, with an extended N-terminal portion that contains various adhesion domains and a well-conserved GPCR autoproteolysis-inducing (GAIN) domain located immediately before the first transmembrane helix[10]. A defining feature of the aGPCR family is that most of the members are autoproteolytically cleaved at a highly conserved GPCR proteolysis site (GPS) in the GAIN domain, which has been suggested to be critical for the maturation and function of these receptors[11]. The cleavage results in two noncovalently associated fragments: an N-terminal fragment (NTF), which includes most of the extracellular domain; and a C-terminal fragment (CTF), which contains a small part of the proteolysed GAIN domain and the transmembrane domain (TMD).

Previous studies have shown that truncating the NTF of some aGPCRs markedly increases signalling[8,12]. It was also implied that the region between the GPS and the TMD—termed the 'stalk'—could function as a tethered agonist for the aGPCRs, as the addition of synthesized stalk peptide increased receptor signalling[8,9]. These findings led to the assumption of a tethered stalk-mediated activation model, including

an inhibitory effect of the NTF on the agonistic activity of the stalk, a dissociation of the stalk peptide from the GAIN domain and a specific interaction between the stalk and TMD that initiates the activation of the receptor[8,9]. In addition, the aGPCRs exhibit notable sequence diversity and lack the conserved activation-related 'micro-switch' motifs that have previously been discovered in class A and class B1 GPCRs[13,14]. These observations suggest that the aGPCRs have a distinct mechanism of signal transduction. However, how the tethered stalk interacts with the TMD and how the activation-required conformational change is relayed from the extracellular surface to the cytoplasmic side remain unknown owing to the lack of an aGPCR structure with the stalk intact, which limits our understanding of the aGPCR signal transduction mechanism that is key for both functional studies and drug discovery.

ADGRD1 (GPR133) and ADGRF1 (GPR110), two representative members of group V and group VI aGPCRs[10], were both recognized as oncogenes in various cancers[15–20]. Both of them are autoproteolytically cleaved and can be activated by the synthetic stalk peptides[8,9], but exhibit sequence variability in some structural motifs that are postulated to be key for class A and B1 receptor activation[13]. To uncover molecular details that govern the tethered agonism of the aGPCRs, we

[1]CAS Key Laboratory of Receptor Research, State Key Laboratory of Drug Research, Shanghai Institute of Materia Medica, Chinese Academy of Sciences, Shanghai, China. [2]University of Chinese Academy of Sciences, Beijing, China. [3]School of Life Science and Technology, ShanghaiTech University, Shanghai, China. [4]iHuman Institute, School of Life Science and Technology, ShanghaiTech University, Shanghai, China. [5]School of Pharmaceutical Science and Technology, Hangzhou Institute for Advanced Study, University of Chinese Academy of Sciences, Hangzhou, China. [6]School of Chinese Materia Medica, Nanjing University of Chinese Medicine, Nanjing, China. [7]Zhongshan Institute for Drug Discovery, Shanghai Institute of Materia Medica, Chinese Academy of Sciences, Zhongshan, China. [8]These authors contributed equally: Xiangli Qu, Na Qiu, Mu Wang, Bingjie Zhang. ✉e-mail: shuiwq@shanghaitech.edu.cn; zhaoq@simm.ac.cn; beiliwu@simm.ac.cn

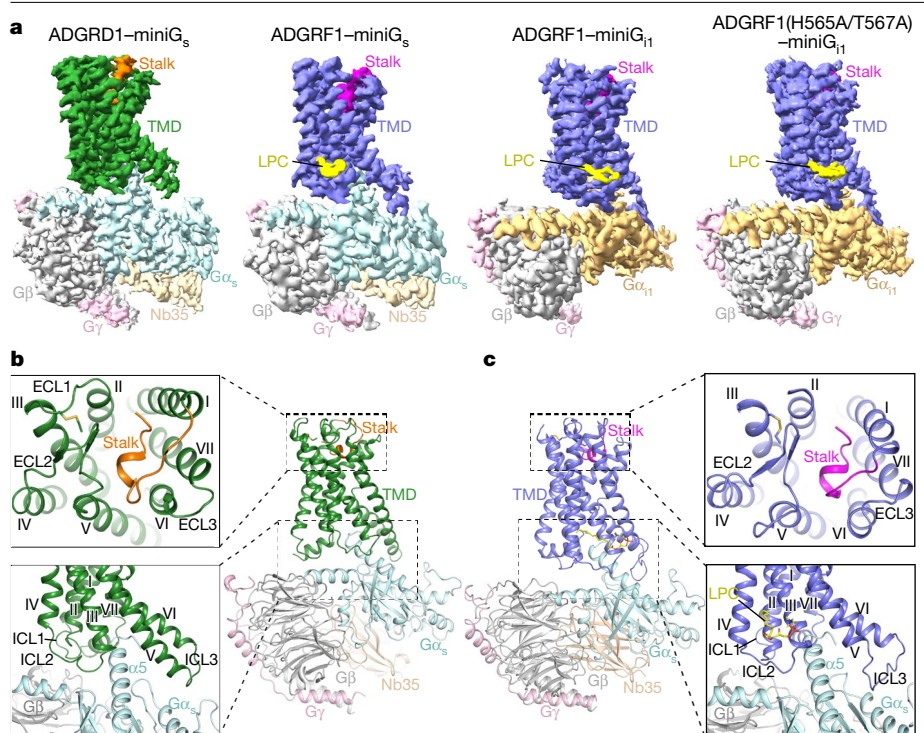

**a** ADGRD1–miniG$_s$   ADGRF1–miniG$_s$   ADGRF1–miniG$_{i1}$   ADGRF1(H565A/T567A)–miniG$_{i1}$

**Fig. 1 | Overall structures of G protein-bound ADGRD1 and ADGRF1.**
**a**, Cryo-EM maps of the ADGRD1–miniG$_s$, ADGRF1–miniG$_s$, ADGRF1–miniG$_{i1}$ and ADGRF1(H565A/T567A)–miniG$_{i1}$ complexes, coloured according to chains. The stalk and TMD of ADGRD1 are coloured orange and green, respectively; the stalk and TMD of ADGRF1 are coloured magenta and blue, respectively; the lipid LPC bound to ADGRF1 is coloured yellow; and Gα$_s$, Gα$_{i1}$, Gβ, Gγ and Nb35 are coloured cyan, gold, grey, pink and light gold, respectively. **b**, Structure of the ADGRD1–miniG$_s$ complex. The structure is shown in cartoon representation. The binding cavities for the stalk and G protein are highlighted by two dashed boxes and are shown in detail on the left. **c**, Structure of the ADGRF1–miniG$_s$ complex. The lipid LPC bound to the receptor intracellular region is shown as yellow sticks. The binding cavities for the stalk and G protein are highlighted by two dashed boxes and are shown in detail on the right.

determined the structures of ADGRD1 and ADGRF1 in complex with different heterotrimeric G proteins. Together with extensive functional studies, the structures reveal many unique features of receptor signal transduction and function modulation.

## Structures of G protein-bound ADGRD1 and ADGRF1

To obtain stable ADGR–G protein complexes, the entire NTF preceding the GPS in ADGRD1 was truncated, whereas for ADGRF1 the GAIN domain was retained (Extended Data Fig. 1a–c). To further optimize protein yield and stability, heterotrimeric G proteins with a shortened Gα subunit (miniGα)[21] were used (Extended Data Fig. 1d, e). Using cryo-electron microscopy (cryo-EM) single-particle analysis, the active structure of ADGRD1 in complex with miniG$_s$ and the structures of ADGRF1 bound to miniG$_s$ or miniG$_{i1}$ were determined (Fig. 1a, Extended Data Figs. 2a–r, 3a–c, Extended Data Table 1). To provide further insights into the autoproteolysis in modulating receptor activation, we also determined the miniG$_{i1}$-bound structure of ADGRF1 with the proteolysis-deficient mutations H565A and T567A introduced in the GPS motif (Fig. 1a, Extended Data Figs. 1a, 2s–x, 3d, Extended Data Table 1). No electron density was observed for the GAIN domain in the cryo-EM maps of all the ADGRF1–G protein structures, suggesting dissociation and/or high dynamics of this region.

Despite poor sequence identity between ADGRD1 and ADGRF1 (28% in the CTFs), the G protein-bound structures of these two aGPCRs exhibit a similar conformation of the CTF with a C$_α$ root mean square deviation (RMSD) of 1.4–1.6 Å (Fig. 1b, c, Extended Data Fig. 4a). The extracellular part of the TMD is in an open 'V' shape with a crevice formed between helices II–V and helices I, VI and VII, which allows the tethered stalk at the N terminus (T545–L558 in ADGRD1, T567–P578 in ADGRF1) to penetrate into the receptor helical bundle (Fig. 1b, c). Comparison with the recently published structure of glucocorticoid–ADGRG3–G$_o$ complex[22] reveals a similar arrangement of helices II–V but large deviations for helices I, VI and VII (Extended Data Fig. 4b). The extracellular ends of helices I, VI and VII in ADGRD1 and ADGRF1 shift clockwise (extracellular view) relative to those in ADGRG3, which produces a large gap between helices I and II to accommodate the N terminus of the stalk. A conformational difference was also observed in the intracellular region

of helices V and VI. In ADGRD1 and ADGRF1, helix VI has a sharp kink that is mediated by the conserved P$^{6.47b}$xxG$^{6.50b}$ motif as a pivot (superscripts refer to the Wootten numbering system for class B GPCRs[23]), a common structural feature shared by the active class B1 secretin GPCRs[24], whereas this helix in ADGRG3 adopts a straight conformation owing to a substitution of P$^{6.47b}$ with S$^{6.47b}$ (Extended Data Fig. 4c). Accompanying this structural deviation, the intracellular tips of helices V and VI in ADGRD1 and ADGRF1 undergo an outward and clockwise movement (intracellular view) by approximate 10 Å compared to those in ADGRG3 (Extended Data Fig. 4d). These conformational differences highlight the diversity of TMD arrangements across the aGPCR family and suggest distinct activation modes of various aGPCRs.

The large conformational difference in helices V and VI results in a more open binding cavity for the G protein in the ADGRD1 and ADGRF1 structures relative to the ADGRG3 structure (Extended Data Fig. 4e). The C terminus of the Gα α5-helix fits into a binding cavity composed of helices II, III, V, VI and VII in ADGRD1 and ADGRF1 (Fig. 1b, c), whereas in ADGRG3 the C terminus of the α5-helix slightly shifts towards helix VII and lacks any contact with helix II (Extended Data Fig. 4f). Similar to what was observed in the ADGRG3 structure, all three intracellular loops of ADGRD1 and ADGRF1 are involved in direct interactions with the G protein, with the first intracellular loop (ICL1) making a close contact with the Gβ subunit and the second and third intracellular loops (ICL2 and ICL3) forming extensive interactions with the Gα subunit (Fig. 1b, c). The G protein-binding modes of ADGRD1 and ADGRF1 are supported by our mutagenesis studies, in which detrimental effects on receptor constitutive activation and/or G protein activation triggered by synthetic stalk peptide (for ADGRD1, T$^{545}$NFAILMQVVPLE$^{557}$ (pD1); for ADGRF1, T$^{567}$SFSILMSPFVP$^{578}$ (pF1); Extended Data Fig. 1f) were observed for some mutations within the G protein-binding pockets of these two aGPCRs (Extended Data Fig. 5a–c, Extended Data Tables 2, 3).

Previous data[9,25] and our own functional studies demonstrate the coupling of ADGRF1 with multiple G proteins, such as G$_s$, G$_q$ and G$_i$ (Extended Data Tables 2, 3). The structures of ADGRF1–miniG$_s$ and ADGRF1–miniG$_{i1}$ provide molecular details of an aGPCR in the recognition of different G protein classes. Similar to what has been observed for the class B1 glucagon receptor GCGR[26], Gα$_s$ and Gα$_i$ share a common binding cavity

on the intracellular surface of ADGRF1 (Extended Data Fig. 4e). The only structural deviation occurs in ICL3 and the intracellular region of helix VI (Extended Data Fig. 4g). To allow accommodation of the bulkier C terminus of $G\alpha_s$, the intracellular tip of helix VI moves outwards by 3 Å (measured at the $C_\alpha$ of $T785^{6.34b}$) in the $miniG_s$-bound structure, which is accompanied by a slight shift of the C terminus of the $G\alpha_s$ $\alpha5$-helix towards helix VI. In association with a longer $\alpha G$–$\alpha 4$ loop in $G\alpha_s$, which causes a steric hindrance, the receptor ICL3 adopts an upward compact structure in the ADGRF1–$miniG_s$ complex but exhibits an extended conformation in the $miniG_{i1}$-bound structure, which results in different patterns of receptor–$G\alpha$ interaction in this region (Extended Data Fig. 4g). This finding attests to the importance of the intracellular loops in governing pleiotropic G protein coupling of the GPCRs.

## The stalk acts as a tethered agonist

The active structures of ADGRD1 and ADGRF1 support the previously proposed activation model of the aGPCRs, in which the stalk region functions as a tethered agonist to activate the receptor[10]. The stalk, which forms a β-strand embedded within a β-sheet core of the GAIN domain in the previously determined crystal structures[27], undergoes a notable conformational rearrangement upon activation. To enable interaction with the TMD, the stalk in ADGRD1 and ADGRF1 exhibits a stacked structure, with its N-terminal half (T545–V553 in ADGRD1; T567–S574 in ADGRF1) lying in a binding cavity within the helical bundle and the C-terminal half adopting an upper position to cap the TMD pocket (Fig. 2a, b). Consistent with the previously observed inhibitory effect of the NTF on receptor activation[8,12,27], the tight binding of the stalk within the GAIN domain constrains its conformational change to block the interaction with the TMD, and thus dissociation from the GAIN is required for the stalk to exert its agonistic activity (Fig. 2c). The autoproteolysis was believed to facilitate the NTF shedding. However, evidence from various studies suggests that the GPS cleavage is not essential for receptor function in vitro and in vivo[8,28]. Indeed, the proteolysis-deficient mutants of ADGRD1 and ADGRF1 exhibited a wild-type level of basal activity in our functional assays (Extended Data Table 2). Of note, a similar stalk–TMD interaction mode was also observed in the $miniG_{i1}$-bound structure of the proteolysis-deficient ADGRF1 (Extended Data Fig. 4h), which shows that the cleavage is not required for the stalk exposure and subsequent stalk-induced receptor activation (Fig. 2d). One possible explanation for this proteolysis-independent activation is that the receptor may exist in multiple conformational states; these are likely to include a portion of receptor molecules in which the stalk is released from the GAIN domain, which leads to a collapse of the original folding of the GAIN. The dissociated stalk tends to interact with the receptor TMD to trigger G protein coupling, which, in turn, stabilizes the stalk–TMD interaction on the extracellular side, and may subsequently induce the stalk exposure of more receptor molecules by altering the equilibrium between different conformational states. An extracellular stimulus that facilitates the stalk exposure may exist, but more evidence is required for a full understanding of this hypothesis.

In all of the G protein-bound structures of ADGRD1 and ADGRF1, the first seven residues in the N-terminal region of the stalk (stalk-N) form a coiled conformation and have a major role in mediating the interaction with the TMD (Fig. 2a, b). This agrees with a previous observation that a core region at the N terminus of the stalk that spans the first 6–8 residues is essential for the agonistic activity[8,9]. Among the aGPCR stalks, the N-terminal residues share strong sequence homology with an aliphatic consensus of TxFAVLM (Extended Data Fig. 6), suggesting a conserved interaction pattern of the stalk binding to the receptor TMD. Indeed, despite the low sequence similarity in the TMD region, ADGRD1 and ADGRF1 accommodate the stalk-N through similar interactions.

The highly conserved stalk residues $F^{S3}$, $L^{S6}$ and $M^{S7}$ (superscripts indicate residue positions in the stalk, abbreviated as 'S') are located at the bottom of the binding cavity with their side chains penetrating deep towards the core of the TMD, forming extensive hydrophobic contacts with helices I, II, III, V, VI and VII and the second extracellular loop (ECL2) (Fig. 2e, g). Alanine substitutions of these three residues abolished the basal activity of ADGRF1 in both cyclic AMP (cAMP) and inositol phosphate accumulation assays, which represents the largest effect among the alanine mutations of the stalk residues (Fig. 2j, Extended Data Table 2). Similarly, the alanine variants $F^{S3}A$, $L^{S6}A$ and $M^{S7}A$ of the stalk-derived peptide pF1 showed a 6–11-fold reduction of agonistic potency in inducing $G_i$ activation of the wild-type ADGRF1 (Extended Data Fig. 5d, Extended Data Table 3). Detrimental effects on receptor basal activity and stalk peptide-induced G protein activation were also observed for the mutations of these three stalk residues in ADGRD1 (Fig. 2i, Extended Data Fig. 5e, Extended Data Tables 2, 3). In addition, mutations of the key TMD residues (mostly conserved in ADGRD1 and ADGRF1; Fig. 2e, g) involved in the interactions with $F^{S3}$, $L^{S6}$ and $M^{S7}$ had notable effects on both constitutive activity and stalk peptide-stimulated activation of ADGRD1 and ADGRF1 (Fig. 2i, j, Extended Data Fig. 5f–h, Extended Data Tables 2, 3). These data provide evidence of the importance of the three stalk residues in receptor activation, which is consistent with previous studies of ADGRG1 and ADGRG6[8,9] and suggests a common tethered stalk-mediated mode of activation of aGPCRs. In the recently published glucocorticoid–ADGRG3–$G_o$ structure[22], the agonist glucocorticoid occupies a binding site similar to that of these stalk residues in the ADGRD1 and ADGRF1 structures (Extended Data Fig. 4i), providing a structural basis for the small-molecule agonist mimicking the tethered agonist to activate the receptor.

In the stalk-N, side chains of the residues at positions S2, S4 and S5 point towards the extracellular milieu, and have a dual role in mediating TMD recognition and cross-talk with the C-terminal region of the stalk (stalk-C) (Fig. 2a, b). These residues form a patch and make contacts with ECL2 and the extracellular tips of helices I, II and VII in ADGRD1 and ADGRF1 (Fig. 2f, h). The importance of this region in mediating receptor activation is reflected by a reduction in basal activity of over 50% for the alanine substitutions of the residues involved in the interactions (Fig. 2i, j, Extended Data Table 2).

In contrast to the extensive interactions contributed by the stalk-N, the stalk-C region (Q552–L558 in ADGRD1, S574–P578 in ADGRF1) forms only limited contacts with the TMD. However, introducing an alanine mutation in this region markedly impaired the receptor basal activity (Fig. 2i, j, Extended Data Table 2). This aligns well with previous investigations of synthetic stalk peptides of several aGPCRs, showing that long peptides with lengths of 12–20 residues exhibit the highest potencies in inducing receptor activation[8,9,29,30]. These data suggest that although the stalk-N confers the agonistic activity, the stalk-C is required for full activity. In the ADGRD1 and ADGRF1 structures, the stalk-C adopts an extended conformation that runs across the helical bundle and packs tightly with the stalk-N, ECL2 and the third extracellular loop (ECL3), largely covering the entrance to the TMD binding pocket. Furthermore, the stalk-C introduces a turn element in the middle of the stalk, which allows the N-terminal tail to bind intramolecularly back toward the binding site. Thus, the requirement of the stalk-C for receptor activation is most likely to result from its contribution to the proper folding of the stalk and stabilization of the stalk-N conformation, which ensure the correct positioning and recognition of the agonistic core sequence in the TMD.

The conserved interactions between the stalk and the TMD that are observed in the G protein-bound structures of ADGRD1 and ADGRF1—especially at the bottom of the binding pocket—suggest that these two aGPCRs may share their agonists to some extent. This was verified by measuring the G protein activation of each receptor using the stalk peptide from the other receptor. We found that pD1 retained its agonistic activity, with only a twofold reduction of half-maximum effective concentration ($EC_{50}$) in inducing ADGRF1 activation compared to pF1, whereas the potency of pF1 was 59-fold lower than that of pD1 when activating ADGRD1 (Extended Data Fig. 5i, j, Extended Data Table 3).

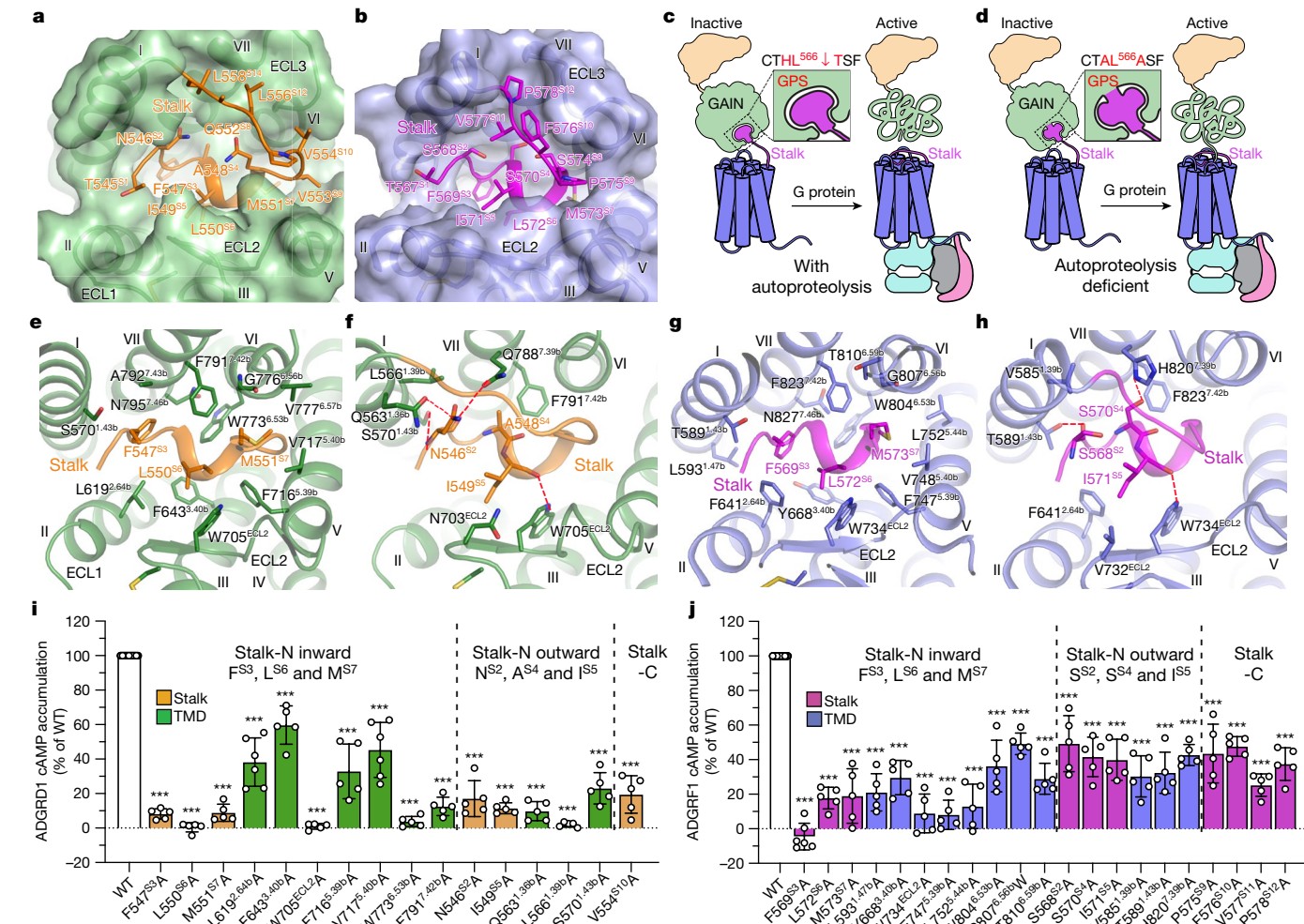

**Fig. 2 | Interaction pattern between the stalk and the TMD. a, b,** The stalk binding cavities in the ADGRD1–miniG_s (**a**) and ADGRF1–miniG_s (**b**) structures. **c,** Schematic diagram of the tethered stalk-mediated activation of ADGRF1 with the autoproteolysis at the GPS. Upon activation, the stalk dissociates from the GAIN domain and then interacts with the TMD. The release of the stalk leads to a collapse of the original folding of the GAIN. **d,** Schematic diagram of the tethered stalk-mediated activation of ADGRF1 with the proteolysis-deficient mutations H565A and T567A introduced in the GPS. The proteolysis is not required for stalk exposure that results in receptor activation and unfolding of the GAIN. **e, g,** Interactions between the TMD and the stalk residues $F^{S3}$, $L^{S6}$ and $M^{S7}$ in ADGRD1 (**e**) and ADGRF1 (**g**). **f, h,** Interactions between the TMD and the stalk residues $N/S^{S2}$, $A/S^{S4}$ and $I^{S5}$ in ADGRD1 (**f**) and ADGRF1 (**h**). Polar interactions are displayed as red dashed lines. **i, j,** Basal activity of wild-type (WT) and mutant versions of ADGRD1 (**i**) and ADGRF1 (**j**), measured by cAMP accumulation assay. The mutants are divided into three groups by dashed lines: (i) mutations of the stalk residues $F^{S3}$, $L^{S6}$ and $M^{S7}$ (stalk-N inward) and the TMD residues that interact with these residues; (ii) mutations of the stalk residues $N/S^{S2}$, $A/S^{S4}$ and $I^{S5}$ (stalk-N outward) and the TMD residues that interact with these residues; and (iii) mutations of the stalk-C residues. Data are presented as a percentage of wild-type activity and are shown as mean ± s.e.m. (bars) from at least five independent experiments performed in technical triplicate with individual data points shown (dots). ***$P < 0.0001$ by one-way analysis of variance followed by Dunnett's post-test compared to the response of wild type. Extended Data Table 2 provides detailed independent experiment numbers ($n$), $P$ values and expression level.

The decreased activity of these stalk peptides relative to the respective peptides of the receptors is probably due to disruption of the interactions in the stalk-C region, which exhibits sequence diversity in the two aGPCRs. Such agonist promiscuity of the stalk-derived peptides was also reported not only within but also between ADGRF and ADGRG subfamilies[29], and may also exist in the other aGPCRs.

## Signalling cascade in ADGRD1 and ADGRF1

The tethered stalk-mediated activation of ADGRD1 and ADGRF1 is achieved by a cooperation of several interaction clusters within the helical bundle as a signalling cascade (Fig. 3a). The stalk initiates signal transduction through a direct interaction with the 'toggle switch' residue $W^{6.53b}$ in these two aGPCRs (Fig. 3a). This highly conserved bulky

residue tethers helices III, V and VI by forming a hydrophobic core with $F^{3.44b}$, $M^{3.47b}$ and $I/V^{5.47b}$ (Fig. 3b, c, Extended Data Fig. 6). As reported for ADGRG3, in which an 'upper quaternary core' in a similar region that mediates helix III–V–VI packing is important for receptor activation[22], in both ADGRD1 and ADGRF1, alanine mutations in this hydrophobic core impaired the receptor basal activity by over 50% and resulted in a substantial reduction of the agonistic potency of the synthetic stalk peptides (Fig. 3h, i, Extended Data Fig. 5k, l, Extended Data Tables 2, 3). The importance of $W^{6.53b}$ is further underlined by its crosstalk with helix VII. The aGPCRs lack the conserved class A $NP^{7.50}xxY$ motif in helix VII (superscript indicates Ballesteros–Weinstein numbering for class A GPCRs[31]), but instead have two highly conserved residues, $Q^{7.49b}$ and $G^{7.50b}$, at a similar position (Extended Data Fig. 6). In the G protein-bound structures of ADGRD1 and ADGRF1, the residue $G^{7.50b}$ introduces a bend

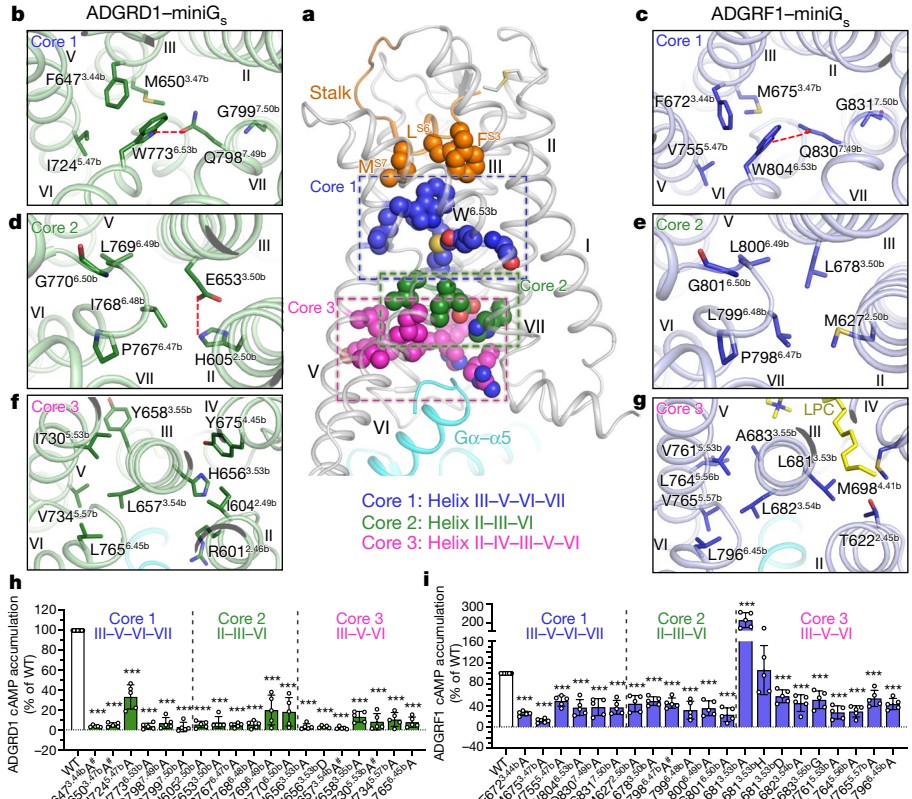

**Fig. 3 | Signalling cascade of ADGRD1 and ADGRF1. a**, Overall view of the interaction cores that are important for receptor activation. The three key interaction cores (cores 1–3) are highlighted by blue, green and magenta dashed boxes, respectively. The stalk residues $F^{S3}$, $L^{S6}$ and $M^{S7}$ are shown as spheres and coloured orange; and the TMD residues in interaction cores 1, 2 and 3 are shown as spheres and coloured blue, green and magenta, respectively. **b**, **d**, **f**, Interactions within cores 1 (**b**), 2 (**d**) and 3 (**f**) in ADGRD1. The residues involved in interactions are shown as green sticks. Polar interactions are shown as red dashed lines. **c**, **e**, **g**, Interactions within cores 1 (**c**), 2 (**e**) and 3 (**g**) in ADGRF1. The residues involved in interactions are shown

as blue sticks. **h**, **i**, Basal activity of wild-type (WT) and mutant versions of ADGRD1 (**h**) and ADGRF1 (**i**), measured by cAMP accumulation assay. Data are presented as a percentage of wild-type activity and are shown as mean ± s.e.m. (bars) from at least five independent experiments performed in technical triplicate with individual data points shown (dots). ***$P < 0.0001$ by one-way analysis of variance followed by Dunnett's post-test compared to the response of wild type. #Low surface expression level (less than 40% of wild-type expression level). Extended Data Table 2 provides detailed independent experiment numbers (*n*), *P* values and expression level.

in the middle of helix VII, which may provide a proper positioning of the intracellular tip of helix VII to assist interaction with the G protein. The residue $W^{6.53b}$ is likely to stabilize the conformation of helix VII by forming a hydrogen bond with the neighbouring residue $Q^{7.49b}$ (Fig. 3b, c). The essential role of this region in modulating receptor function is reflected by a considerable impairment of both the receptor basal activity and the stalk peptide-induced G protein activation associated with the mutations $Q^{7.49b}A$ and $G^{7.50b}A$ in ADGRD1 and ADGRF1 (Fig. 3h, i, Extended Data Fig. 5k, l, Extended Data Tables 2, 3). The above data suggest that the interaction network involving $W^{6.53b}$ and helices III, V and VII underneath the stalk binding pocket has a crucial role in sensing the stalk binding and stabilizing the receptor in an active state.

In a lower region towards the intracellular side, the active structures of ADGRD1 and ADGRF1 are further stabilized by an interaction core composed of four residues at positions 2.50b, 3.50b, 6.48b and 6.49b in the centre of the helical bundle (Fig. 3a, d, e). Formation of this interaction core is facilitated by the sharp kink at the $P^{6.47b}xxG^{6.50b}$ motif of helix VI, which allows helix VI to approach helices II and III (Fig. 3d, e). Lacking a bend in helix VI, such an inter-helix interface does not exist in the active ADGRG3, in which the aliphatic chain of a palmitoylation attached to the $G_o$ protein tethers the transmembrane helices in this region[22] (Extended Data Fig. 4c). Further mutagenesis studies underline the requirement of the helix-VI kink for receptor activation of ADGRD1 and ADGRF1, as mutating $P^{6.47b}$ or $G^{6.50b}$ in both receptors led to a marked loss of receptor constitutive activity and synthetic stalk peptide

potency (Fig. 3h, i, Extended Data Fig. 5m, n, Extended Data Tables 2, 3). The $P^{6.47b}xxG^{6.50b}$ motif is highly conserved in class B1 secretin receptors but only present in the aGPCR subfamilies of ADGRB, ADGRD and ADGRF (Extended Data Fig. 6), suggesting that these aGPCRs may share a common bended conformation of helix VI that probably results in a similar helix II–III–VI packing interaction core. However, the residues within the core exhibit sequence variability between receptors (Extended Data Fig. 6). Although the positions 6.48b and 6.49b are conserved with two hydrophobic residues in both ADGRD1 and ADGRF1, the residues at 2.50b and 3.50b are charged in ADGRD1 ($H605^{2.50b}$ and $E653^{3.50b}$) but apolar in ADGRF1 ($M627^{2.50b}$ and $L678^{3.50b}$). Thus, the helix II–III–VI interaction core in ADGRF1 is associated purely through hydrophobic contacts, whereas in ADGRD1 $H605^{2.50b}$ and $E653^{3.50b}$ form an extra salt bridge (Fig. 3d, e). Alanine substitutions in this core region—which disrupt the interaction patch and probably destabilize the helical bundle—had a detrimental effect on receptor activity (Fig. 3h, i, Extended Data Fig. 5m, n, Extended Data Tables 2, 3).

In aGPCRs, the residues at positions equivalent to the highly conserved and functionally important class A $D/ER^{3.50}Y$ motif exhibit a notable diversity in sequence (Extended Data Fig. 6). In ADGRD1 and ADGRG3, the residues are replaced with an $HL^{3.54b}Y$ motif, whereas a hydrophobic sequence $LL^{3.54b}A$ is found in ADGRF1. Despite the distinct sequences, this motif has a major role in shaping the intracellular binding interface for the G protein. In addition to its interaction with the G protein, this motif makes extensive contacts with helices II, IV,

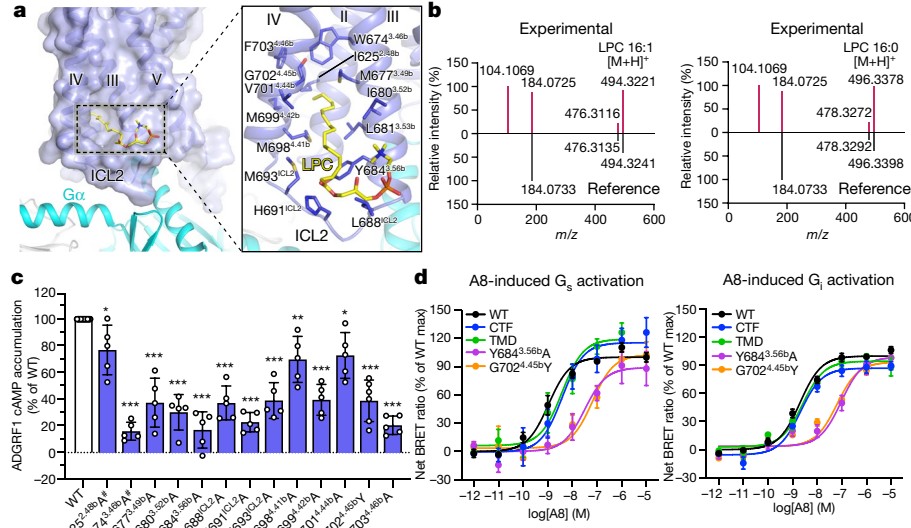

**Fig. 4 | Lipid molecule in ADGRF1. a**, Lipid-binding site in ADGRF1. The ADGRF1–miniG$_s$ structure is shown in cartoon representation. The receptor is also shown as surface. The lipid LPC is shown as yellow sticks. The receptor residues involved in lipid binding are shown as blue sticks. **b**, High-resolution tandem mass spectrometry (MS/MS) spectra of two LPC molecules specifically associated with ADGRF1. Their experimental spectra matched with the reference spectra recorded in the Lipid-Blast database. **c**, Basal activity of wild-type (WT) and mutant versions of ADGRF1, measured by cAMP accumulation assay. Data are presented as a percentage of wild-type activity and are shown as mean ± s.e.m. (bars) from at least five independent

experiments performed in technical triplicate with individual data points shown (dots). *$P < 0.05$, **$P < 0.001$, ***$P < 0.0001$ by one-way analysis of variance followed by Dunnett's post-test compared to the response of wild type. #Low surface expression level (less than 40% of wild-type expression level). Extended Data Table 2 provides detailed independent experiment numbers (*n*), *P* values and expression level. **d**, A8-induced G$_s$ and G$_i$ activation of ADGRF1. Data are shown as mean ± s.e.m. from at least four independent experiments performed in technical duplicate. Extended Data Table 3 provides detailed independent experiment numbers (*n*), *P* values, statistical evaluation and expression level.

V and VI, which greatly stabilizes the conformation of the receptor intracellular region (Fig. 3f, g). The residue at position 3.53b points towards helices II and IV in both ADGRD1 and ADGRF1, but its alanine mutation had different effects on receptor activation. Similar to what was observed for ADGRG3[22], the H656[3.53b]A mutation of ADGRD1 abolished receptor basal activity and suppressed pD1 potency by 79-fold. However, the ADGRF1 mutant L681[3.53b]A retained the wild-type activity (Fig. 3h, i, Extended Data Fig. 5o, p, Extended Data Tables 2, 3). This may be explained by different interaction environments of this residue in the active structures of these two aGPCRs. In ADGRD1, the bulky side chain of H656[3.53b] is required for making contacts with the neighbouring helices (Fig. 3f), whereas for ADGRF1, the association between helices II, III and IV in this region is also mediated by a lipid molecule as discussed below (Fig. 3g). Thus, removing the side chain is more detrimental to ADGRD1 activation. We also tested different charges at this position. The results showed that histidine was also allowed at this position in ADGRF1, but an aspartic acid substitution strongly impaired the activation of both receptors (Fig. 3h, i, Extended Data Fig. 5o, p, Extended Data Tables 2, 3), suggesting that a positive charge here is more beneficial than a negative charge—a feature different from class A GPCRs. One of the possible explanations for this is that a negatively charged residue may hinder the conformational change of the receptor from an inactive state to an active state, which would require an inactive aGPCR structure for full understanding.

Compared to the residue at position 3.53b, the other two residues 3.54b and 3.55b in this motif are relatively conserved, with hydrophobic amino acids in most aGPCRs (Extended Data Fig. 6). These residues in ADGRD1 and ADGRF1 build a hydrophobic interaction network with a hydrophobic patch in helices V and VI, including I/V[5.53b], L[5.56b] (only in ADGRF1), V[5.57b] and L[6.45b], and residues Y/C[G.H5.23] and L[G.H5.25] at the C terminus of Gα (superscripts refer to the common Gα numbering system[32]) (Fig. 3f, g). The importance of this interaction cluster is reflected by a loss of receptor basal activity of more than 40% and a 4–103-fold decrease in stalk peptide potency for the alanine or glycine

replacements of the receptor residues within the cluster (Fig. 3h, i, Extended Data Fig. 5o–r, Extended Data Tables 2, 3).

## Lipid-regulated activation of ADGRF1

Previous structural studies of GPCRs revealed the involvement of lipid molecules in function modulation for several different receptors including the recently published ADGRG3, in which the G protein-attached palmitoylation inserts deep into the receptor core[22]. A similar binding mode of palmitoylation is excluded in ADGRD1 and ADGRF1 owing to a steric hindrance caused by the sharp kink in helix VI (Extended Data Fig. 4c). Alternatively, the cryo-EM maps of all three G protein-bound ADGRF1 structures display strong densities for a lipid molecule bound to the intracellular region of the receptor (Figs. 1a, 4a). We then performed lipidomics analysis to identify putative lipids associated with the receptor. Using ADGRD1 as a control, we discovered that two lysophosphatidylcholine (LPC) molecules—LPC 16:0 and LPC 16:1—bound specifically to ADGRF1, and that other classes of phospholipids did not (Fig. 4b, Extended Data Fig. 7). The identified LPC lipid, which contains a phosphocholine head group and a long fatty-acyl chain of 16 carbons, fits perfectly into the cryo-EM maps (Extended Data Figs. 1g, 3b–d). It stretches from the membrane lipid bilayer to the intracellular tip of helix II, with its fatty acyl chain penetrating into a 'tunnel' shaped by ICL2 and helices III and IV, forming extensive hydrophobic contacts with the receptor (Fig. 4a). To the best of our knowledge, this is the first case of an LPC ligand being associated with a GPCR, although previous studies have reported the binding of phosphatidylcholine, phosphatidylethanolamine or phosphatidylinositol to certain receptors[33,34].

The LPC molecule serves as an anchor of the receptor intracellular region, stabilizing the conformations of ICL2 and the intracellular ends of helices II and III, which are functionally important and have extensive interactions with the G protein. Thus, the lipid may have a role in stabilizing the receptor in the active state. This is supported by a notable reduction of receptor basal activity associated with the

alanine substitutions of most of the residues in the lipid-binding site (Fig. 4c, Extended Data Table 2). Furthermore, a marked impairment of receptor constitutive activation was also observed for the G702$^{4.45b}$Y mutant, which is expected to form a severe clash with the terminus of the fatty acyl chain to repel the lipid binding (Fig. 4a, c, Extended Data Table 2). Among the aGPCRs, the residue G$^{4.45b}$ only exists in ADGRF1 and ADGRF4, whereas in the other receptors the counterparts have long side chains (Y, F, M, L or I) that probably block lipid binding (Extended Data Fig. 6). Consistent with this, no lipid was found in this region of the active ADGRD1, which has a bulky tyrosine residue at this position. Therefore, it is likely that this lipid-binding site is unique to ADGRF1 and potentially ADGRF4. These findings from the aGPCR structures highlight the importance of lipid molecules in the modulation of receptor function and the diversity of modes of action of the lipids.

*N*-docosahexaenoylethanolamine (synaptamide), a synaptogenic metabolite of docosahexaenoic acid (Extended Data Fig. 1g), promotes neurogenesis, neuritogenesis and synaptogenesis, and has been reported as an endogenous small-molecule agonist for ADGRF1[35]. It activates the receptor in a stalk-independent manner and was believed to trigger signalling through an interaction with the extracellular GAIN domain[36]. To further study the behaviour of lipid molecules in modulating receptor function, we measured both $G_s$ and $G_i$ activation of ADGRF1 induced by A8, a methylated analogue of synaptamide[37] (Extended Data Fig. 1g). Notably, when the GAIN domain was removed, the CTF or TMD retained the wild-type level of A8 potency (Fig. 4d, Extended Data Table 3). These data strongly suggest that A8 exerts its agonistic activity by specifically binding to the TMD of ADGRF1. This raised another question of whether this lipid molecule recognizes the same lipid-binding site as that observed in the active ADGRF1 structures. Thus, we further tested the effect of two mutations in the lipid-binding site—Y684$^{3.56b}$A and G702$^{4.45b}$Y—on the A8-induced receptor activation, and the results showed a 30–60-fold reduction of the A8 potency (Fig. 4d, Extended Data Table 3). On the basis of these data, we suspect that synaptamide binds to the intracellular lipid-binding site, and may activate the receptor by triggering a conformational rearrangement of the receptor intracellular region.

In summary, this work provides structural and mechanistic insights into the tethered stalk-mediated activation of ADGRD1 and ADGRF1. The activation is initiated by extensive interactions between the stalk and the TMD, facilitated by a cascade of inter-helix interaction cores, and further modulated by a lipid molecule that specifically binds to the receptor intracellular region. These features have not to our knowledge been observed in any other GPCR structures that have been reported so far, and thus greatly extend our understanding of GPCR signalling.

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

## Methods

### Construct cloning

To enable structure determination, the NTF preceding the GPS (residues M1–L544) and the C-terminal residues S828–V874 in ADGRD1 were truncated. For ADGRF1, the N-terminal region preceding the GAIN domain (residues M1–R250) and the flexible C terminus (residues Q861–E910) were removed. To generate the proteolysis-deficient ADGRF1, two mutations, H565A and T567A, were introduced in the GPS motif of the truncated receptor. The codon-optimized genes of human ADGRD1 (Uniprot number: Q6QNK2-1; residues T545–T827) and ADGRF1 (Uniprot number: Q5T601-1; residues V251–K860) were cloned into a modified pFastBac1 vector (Invitrogen) with a haemagglutinin (HA) signal peptide at the N terminus. To facilitate expression and purification, a Flag epitope tag and a twin-strep tag were added to the C terminus of ADGRD1, whereas for ADGRF1 the Flag and strep tags were added to the N and C termini of the receptor, respectively. To improve protein yield and stability, heterotrimeric G proteins with a shortened Gα subunit (miniGα)[21], which lacks the α-helical domain, were used to form complexes with ADGRD1 and ADGRF1. Dominant-negative miniGα subunits were generated by introducing several mutations (miniGα$_s$, G49D, E50N, A249D, S252D, L272D, I372A and V375I; miniGα$_i$, G42D, E43N, G217D, T219A, A226D, P287Q, V332A and V335I) to further improve the stability of the heterotrimeric G protein complexes[38]. The genes of miniGα$_s$ and miniGα$_i$ were cloned into the pFastBac1 vector with a 6×His tag adding to their N termini. The genes of human Gβ$_1$ with an N-terminal 6×His tag and Gγ$_2$ were subcloned into a pFastBac Dual vector (Invitrogen). All mutants used for structural and functional studies were generated by using site-directed mutagenesis PCR.

### Expression and purification of the G protein-bound ADGRD1 and ADGRF1 complexes

The G protein-bound ADGRD1 and ADGRF1 complexes were obtained by co-expressing the receptor, miniGα and Gβ$_1$γ$_2$ in High Five insect cells (Invitrogen). The cells were routinely tested for mycoplasma contamination. The high titre recombinant viral stocks were generated using a Bac-to-Bac Baculovirus Expression System (Invitrogen) and were used to transfect the insect cells at a density of $1.5 \times 10^6$ cells per ml with a multiplicity of infection ratio of 1:1:1. The transfected cells were further cultured at 27 °C for 48 h before collection.

The cells expressing the ADGRD1- or ADGRF1–G protein complexes were collected by centrifugation and suspended in a buffer containing 20 mM HEPES, pH 7.5, 50 mM NaCl, 2 mM MgCl$_2$ and EDTA-free protease inhibitor cocktail tablets (Roche) using dounce homogenization. The suspended membrane solution was supplemented with 25 mU ml$^{-1}$ apyrase and incubated at room temperature for 2 h. The membrane pellets were collected by centrifugation at 20,000$g$ for 30 min. The complex proteins were then extracted from the membranes by incubating with a solubilization buffer containing 20 mM HEPES, pH 7.5, 150 mM NaCl, 2 mM MgCl$_2$, 0.5% (w/v) lauryl maltose neopentyl glycol (LMNG, Anatrace) and 0.05% (w/v) cholesteryl hemisuccinate (CHS, Anatrace) at 4 °C for 2 h. The supernatant was collected by centrifugation at 30,000$g$ for 30 min and incubated with Strep-Tactin XT Sepharose resin (IBA Lifesciences) overnight at 4 °C. For the ADGRD1–miniG$_s$ and ADGRF1–miniG$_s$ complexes, a 1.5 molar excess of nanobody35 (Nb35; see below for protocols of expression and purification) was added at the beginning of this incubation process to improve complex stability.

The resin was collected by centrifugation at 800$g$ for 5 min and washed with 4 column volumes of 20 mM HEPES, pH 7.5, 150 mM NaCl, 2 mM MgCl$_2$, 0.01% (w/v) LMNG and 0.001% (w/v) CHS to decrease the LMNG concentration. Detergent exchange was performed by incubating the resin with 20 mM HEPES, pH 7.5, 150 mM NaCl, 2 mM MgCl$_2$ and 0.25% (w/v) glyco-diosgenin (GDN, Anatrace) at 4 °C for 2 h. The resin was then washed with 10 column volumes of 20 mM HEPES, pH 7.5, 150 mM NaCl, 2 mM MgCl$_2$ and 0.01% (w/v) GDN. The complex

protein was eluted with 5 column volumes of 200 mM Tris-HCl, pH 8.0, 150 mM NaCl, 2 mM MgCl$_2$, 0.01% (w/v) GDN and 50 mM biotin, and further incubated with Ni-NTA resin (Clontech) at 4 °C for 1 h. The resin was collected and washed with 10 column volumes of 20 mM HEPES, pH 7.5, 150 mM NaCl, 2 mM MgCl$_2$ and 0.01% (w/v) GDN. The complex protein was then eluted with the same buffer supplemented with 300 mM imidazole and loaded to size-exclusion chromatography (SEC) using a Superdex 200 Increase 10/300 GL column (GE Healthcare) pre-equilibrated with 20 mM HEPES, pH 7.5, 150 mM NaCl, 2 mM MgCl$_2$ and 0.01% (w/v) GDN. The complex fractions were pooled and concentrated to 3 mg ml$^{-1}$ using a 100-kDa molecular weight cut-off concentrator (Millipore). Protein purity and homogeneity were analysed using SDS–PAGE and analytical SEC.

### Expression and purification of Nb35

Nb35 was expressed and purified as previously described with modifications[26]. In brief, the C-terminal 6×His-tagged Nb35 gene was cloned into a pET28a vector and expressed in *Escherichia coli* stain BL21 (DE3). The cells were cultured in LB medium supplemented with 50 μg ml$^{-1}$ kanamycin at 37 °C until reaching an optical density at 600 nm (OD$_{600 nm}$) of 0.6. After adding 1 mM IPTG, the cultures were then grown at 16 °C for 12 h. The cell pellets were collected by centrifugation at 4,000$g$ for 30 min and then lysed in 10 mM HEPES, pH 7.5 and 100 mM NaCl by sonication. The supernatant was isolated by centrifugation at 30,000$g$ for 30 min, and incubated with Ni-NTA resin at 4 °C for 1 h. The resin was then washed with 20 column volumes of 10 mM HEPES, pH 7.5, 100 mM NaCl and 30 mM imidazole. The Nb35 protein was eluted with 10 column volumes of 10 mM HEPES, pH 7.5, 100 mM NaCl and 300 mM imidazole, and further purified by SEC using a Superdex 75 10/300 GL column (GE Healthcare) pre-equilibrated with 10 mM HEPES, pH 7.5 and 100 mM NaCl. Peak fractions were pooled together and concentrated to 3 mg ml$^{-1}$. The final Nb35 sample was supplemented with 10% glycerol and stored at −80 °C until use.

### Cryo-EM data acquisition

The formation of ADGRD1- and ADGRF1–G protein complexes was confirmed by negative staining electron microscopy and the sample quality was evaluated by a 200 kV Talos Arctica G2 electron microscope (FEI). For data acquisition, 3 μl of purified complex sample was applied to glow-discharged 300-mesh gold grids (CryoMatrix M024-Au300-R12/13) and followed by vitrification via plunge-freezing in liquid ethane cooled by liquid nitrogen using Vitrobot Mark IV (Thermo Fisher Scientific) with 1.5 s blot time and 0 blot force at 4 °C and 100% humidity. The well-prepared grids were selected for data acquisition by using a 300 kV Titan Krios G3 electron microscope (FEI) equipped with a K3 Summit direct electron detector (Gatan) at a nominal magnification of 81,000× and a GIF-Quantum LS Imaging energy filter with a slit width of 20 eV. Images were captured by SerialEM[39] with a physical pixel size of 1.071 Å and a defocus ranging from −0.8 μm to −1.5 μm. Each image stack comprised 40 frames in a total of 3 s with 0.075 s exposure per frame, and the total dose was 70 electrons per Å$^2$.

### Cryo-EM data processing and map construction

The image stacks of the ADGRD1- and ADGRF1–G protein complexes were subjected to beam-induced motion correction by MotionCor2[40]. Contrast transfer function (CTF) parameters for each image were determined by Gctf v.1.18[41]. The particle projections were extracted by template-free auto-picking of RELION 3.1[42]. Two-dimensional (2D) classification, three-dimensional (3D) classification, 3D auto-refinement, Bayesian polishing and CTF refinement were performed using RELION 3.1. The resolution of density maps was calculated by the gold-standard Fourier shell correlation (FSC) with the 0.143 criterion. After sharpening by post-processing in RELION 3.1, ResMap v.1.1.4 was used to estimate the local resolution[43].

For the ADGRD1–miniG$_s$ complex, a total of 4,588 movies were collected and subjected to beam-induced motion correction and CTF

determination. A total of 3,307,950 particle projections were produced by reference-free auto-picking and subjected to two rounds of 2D classification to discard false-positive particles. An ab initio model generated by RELION 3.1 was used as an initial reference model for 3D classification. A subset of 3,195,673 particles was selected for another round of 3D classification. The best-looking dataset of 1,266,674 particles was subjected to CTF refinement, Bayesian polishing and 3D auto-refinement, resulting in a final map at 2.8 Å resolution.

For the ADGRF1–miniG$_{i1}$ complex, a total of 14,521 movies were collected and processed separately as three datasets of 3,031, 6,921 and 4,569 movies. All datasets were submitted to beam-induced motion correction and CTF determination. A total of 3,781,704, 8,302,989 and 5,610,993 particle projections were respectively extracted by reference-free auto-picking and subjected to 2D classification to discard false-positive particles. An ab initio model generated by RELION 3.1 was used as an initial reference model for 3D classification. The best model was selected as the reference model for another round of 3D classification. The best-looking classes from the three datasets were subjected to CTF refinement and Bayesian polishing, and then combined for 3D auto-refinement and another round of focused 3D classification with a mask over the receptor–G protein complex. A dataset of 1,735,602 particles from the focused 3D classification was subjected to another round of 3D auto-refinement, generating a map with a global resolution of 3.4 Å.

A total of 10,299 movies of ADGRF1–miniG$_s$ were collected and subjected to beam-induced motion correction and CTF determination. A total of 6,972,863 particle projections were extracted by reference-free auto-picking and subjected to three rounds of 2D classification to discard false-positive particles. The model of ADGRF1–miniG$_{i1}$ complex was low-passed to 60 Å and used as an initial reference model for 3D classification. The best model was selected as the reference model for another two rounds of 3D classification. The best-looking class with 365,932 particles was selected and subjected to CTF refinement, Bayesian polishing and 3D auto-refinement, resulting in a map with a global resolution of 3.1 Å.

A total of 9,125 movies of the ADGRF1(H565A/T567A)–miniG$_{i1}$ complex were collected and subjected to beam-induced motion correction and CTF determination. A total of 9,258,154 particle projections were extracted by reference-free auto-picking and subjected to 2D classification to discard false-positive particles. An ab initio model generated by RELION 3.1 was used as a reference model for 3D classification. The best-looking classes of 799,431 particles were subjected to CTF refinement, Bayesian polishing and 3D auto-refinement, resulting in a map with a global resolution of 3.0 Å.

### Model building and refinement

The models of the ADGRD1– and ADGRF1–G protein complexes were built by recruitment of the receptors from AlphaFold predicted models[44], the subunits of Gα$_i$, Gβ and Gγ from the glucagon–GCGR–G$_i$ structure (Protein Data Bank (PDB) ID: 6LML), and the Gα$_s$ and Nb35 from the glucagon–GCGR–G$_s$ structure (PDB: 6LMK) as initial templates. Each model was docked into the corresponding cryo-EM density map by ChimeraX v.1.1[45], followed by iterative manual adjustment in Coot[46] and real-space refinement in phenix.real_space_refine of PHENIX[47]. The model statistics were validated using MolProbity[48].

The final model of ADGRD1–miniG$_s$ contains 277 residues of ADGRD1 (T545–T821), 210 residues of miniGα$_s$ (I26–K58, F208–N254 and R265–L394), 339 residues of Gβ$_1$ (S2–N340), 56 residues of Gγ$_2$ (A7–R62) and 128 residues of Nb35 (Q1–S128). The final ADGRF1–miniG$_s$ model contains 280 residues of ADGRF1 (T567–V647 and S654–K852), 211 residues of miniGα$_s$ (I26–K58, I207–N254 and R265–L394), 339 residues of Gβ$_1$ (S2–N340), 56 residues of Gγ$_2$ (A7–R62) and 127 residues of Nb35 (Q1–S127). For the ADGRF1–miniG$_{i1}$ complex, the final model contains 286 residues of ADGRF1 (T567–K852), 207 residues of miniGα$_i$ (K10–M53, T182–Y230 and N241–F354), 339 residues of

Gβ$_1$ (S2–N340) and 56 residues of Gγ$_2$ (A7–R62). For the ADGRF1(H565A/T567A)–miniG$_{i1}$ complex, the final model contains 286 residues of ADGRF1 (A567–K852), 207 residues of miniGα$_i$ (K10–M53, T182–Y230, N241–F354), 339 residues of Gβ$_1$ (S2–N340) and 56 residues of Gγ$_2$ (A7–R62). The remaining residues of the receptors and G proteins are disordered and were not modelled. The final refinement statistics are provided in Extended Data Table 1. The overfitting during refinement was excluded by refining the final model against one of the half maps and by comparing the resulting map versus model FSC curves with the two half maps and the final model. The structure figures were prepared using PyMOL v.1.8 and UCSF Chimera v.1.15.

### cAMP accumulation assay

The wild-type ADGRD1 and ADGRF1 and mutants used in functional studies were constructed into a pTT5 vector with a Flag tag at the N terminus for receptor expression measurement. The basal activity of ADGRD1 and ADGRF1 in mediating G$_s$ signalling was measured by a cAMP accumulation assay using a LANCE Ultra cAMP detection kit (PerkinElmer) following the manufacturer's instruction. In brief, 2 ml HEK293F cells (Invitrogen; cells were routinely tested for mycoplasma contamination) at a density of $1.2 \times 10^6$ cells per ml were transiently transfected with 2,000 ng plasmid of the wild-type or mutant receptor and cultured at 37 °C for 48 h with 5% $CO_2$ atmosphere in a shaker shaking at 220 rpm. After collection, the cell-surface expression of the receptors was measured by incubating 10 µl cells with 15 µl monoclonal anti-Flag M2-FITC antibody (Sigma; 1:120 diluted in TBS supplemented with 4% BSA and 20% viability staining solution 7-AAD (Invitrogen)) at 4 °C for 20 min. After incubation, 175 µl TBS buffer was added and the fluorescent signal was measured using a flow cytometry reader (Guava easyCyte HT, Millipore).

Ten microlitres of cells were dispensed into 384-well plates (6,000 cells per well suspended in stimulation buffer (HBSS buffer (Thermo Fisher Scientific) supplemented with 5 mM HEPES, pH 7.4, 0.1% BSA (PerkinElmer) and 0.5 mM IBMX (Sigma)), incubated at room temperature for 30 min and then treated with 5 µl Eu-cAMP tracer and 5 µl ULight-anti-cAMP working solution at room temperature for 1 h. Fluorescent signals were acquired by a Synergy II (Bio-Tek) plate reader with excitation at 330 nm and emission at 620 nm and 665 nm. The cAMP accumulation was calculated by a standard dose–response curve using GraphPad Prism 8.0.

### Inositol phosphate accumulation assay

An inositol monophosphate (IP1) accumulation assay was performed to measure the basal activity of ADGRF1 in mediating G$_q$ signalling by using an IP-One Gq assay kit (Cisbio Bioassays) following the manufacturer's instructions. The wild-type ADGRF1 and mutants were expressed in HEK293F cells and the expression levels were measured as described above. Fourteen microlitres of cells were dispensed into 384-well plates (18,000 cells per well suspended in stimulation buffer) and incubated at 37 °C for 1.5 h. Then 3 µl IP1-d2 antibody (1:20 diluted in lysis and detection buffer) and 3 µl cryptate-labelled anti-IP1 monoclonal antibody (1:20 diluted in lysis and detection buffer) were added and incubated at room temperature for 1 h. Fluorescent signals were measured by the Synergy II (Bio-Tek) plate reader with excitation at 330 nm and emission at 620 nm and 665 nm. The accumulation of IP1 was calculated according to a standard dose–response curve using GraphPad Prism v.8.0.

### BRET assay using TRUPATH biosensors

To study the synthetic stalk peptide-induced G protein activation of ADGRD1 and ADGRF1, a BRET assay using TRUPATH biosensors was conducted to measure the proximal interaction between RLuc8 fused to the Gα subunit and GFP2 fused to the Gγ subunit. The TRUPATH suite of biosensors was obtained from Addgene (Addgene kit no. 1000000163) as a gift from B. Roth, including Gα$_{sS}$-RLuc8, Gα$_{i1}$-RLuc8, Gβ$_3$ and Gγ$_9$-GFP2 as previously described[49]. The stalk peptides pD1 and

pF1 were synthesized (GL Biochem), dissolved in dimethyl sulfoxide (DMSO) at a concentration of 50 mM as stock solutions, and diluted to different concentrations with assay buffer (HBSS buffer (Thermo Fisher Scientific) supplemented with 20 mM HEPES, pH 7.4) upon assay.

The wild-type or mutant ADGRD1 and ADGRF1 were transiently co-transfected with plasmids encoding Gα-RLuc8 (Gα$_{ss}$-RLuc8 for G$_s$ activation assay; Gα$_{i1}$-RLuc8 for G$_i$ activation assay), Gβ$_3$ and Gγ$_9$-GFP2 at a ratio of 2:1:1:1 (receptor plasmid, 800 ng; G protein subunit plasmids, 400 ng for each) in 2 ml HEK293F cells at a density of $1.2 \times 10^6$ cells per ml. Cell cultivation and receptor surface expression measurement were performed as described above. The cells were plated into 96-well white plates (40,000 cells per well) in 60 μl of assay buffer and incubated at 37 °C for 30 min. Then 10 μl of freshly prepared 50 μM coelenterazine 400a (Nanolight Technologies) was added. After equilibration for 5–10 min, the BRET baselines were measured by the Synergy II (Bio-Tek) plate reader with 410 nm (RLuc8-coelenterazine 400a) and 515 nm (GFP2) emission filters for 15 min. The cells were stimulated with 30 μl of synthetic stalk peptide at different concentrations and the BRET signals were monitored continuously five times. The last measurements were used in data analysis. The BRET ratios were calculated as the ratio of the GFP2 emission to RLuc8 emission.

## Identification of phospholipid ligands by LC–MS/MS

The identification of phospholipids specifically bound to ADGRF1 was performed as previously described[33,50] with minor modifications. In brief, ADGRF1 and control ADGRD1 protein were reduced with 5 mM TCEP at 25 °C for 30 min and alkylated with 20 mM idoacetamide at 25 °C for 30 min. Then the protein samples were digested with trypsin (Promega) at an enzyme-to-protein ratio of 1:50 (w/w) at 37 °C overnight. The digested proteins were dried in a vacuum concentrator and then extracted with 400 μl of ice-cold methanol:water (9:1, v/v) by vortex and sonication. After centrifugation at 12,000$g$ for 15 min at 4 °C, the supernatants were collected and lyophilized under vacuum. The lipid extracts were resuspended in methanol:chloroform (9:1, v/v) to an equivalent concentration of 2 μM. Samples were analysed on a Q Exactive mass spectrometer (Thermo Fisher Scientific) operating in the positive ion mode coupled to a Waters Acquity UPLC system (Waters). The liquid chromatography (LC) separation was performed on a CSH C18 column (100 × 2.1 mm; 1.7 μm) (Waters) at a flow rate of 0.4 ml min$^{-1}$ at 40 °C, with the mobile phase A consisting of acetonitrile:water (60:40, v/v) with 10 mM ammonium formate and 0.1% formic acid, and B consisting of 2-propanol:acetonitrile (90:10, v/v) with 10 mM ammonium formate and 0.1% formic acid. The LC gradient was set as follows: 0 min 15% B; 0–4 min 30% B; 4–4.5 min 48% B; 4.5–22 min 82% B; 22–24 min 99% B; 24–30 min 15% B. The acquisition method was set to the following parameters: mass range 100–1,500 $m/z$; spray voltage 3.5 kV; sheath gas (nitrogen) flow rate 35 units; auxiliary gas (nitrogen) flow rate 10 units; capillary temperature 320 °C. MS1 scan parameters included resolution 70,000, AGC target 3e6, and maximum injection time 200 ms. MS/MS spectra were acquired on the top 10 precursors with collision energy set at 20 eV. All samples were prepared in three independent replicates.

Phospholipids in the ADGRF1 and control samples were identified in MS-DIAL (v.4.70) by matching accurate mass and tandem mass spectra with a built-in lipid spectral library Lipid-Blast[51]. Then the extracting ion chromatograms (EICs) of identified lipids were acquired from each sample using TraceFinder (v.4.0, Thermo Fisher Scientific) based on accurate mass matching and retention time alignment with respective peaks. The specificity of lipid binding to ADGRF1 was assessed by the ratio of EIC peak areas for each lipid in the ADGRF1 versus the control sample. Lipids with a mean EIC ratio > 2 and $P < 0.05$ ($n = 3$) were defined as specific binders to ADGRF1[52].

## Reporting summary

Further information on research design is available in the Nature Research Reporting Summary linked to this paper.

## Data availability

Atomic coordinates and cryo-EM density maps for the structures of ADGRD1–miniG$_s$, ADGRF1–miniG$_s$, ADGRF1–miniG$_{i1}$ and ADGRF1(H565A/T567A)–miniG$_{i1}$ complexes have been deposited in the PDB under identification codes 7WU2, 7WU3, 7WU4 and 7WU5, respectively, and in the Electron Microscopy Data Bank under accession codes EMD-32817, EMD-32818, EMD-32819 and EMD-32820, respectively.

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

**Acknowledgements** The cryo-EM studies were performed at the electron microscopy facility of Shanghai Institute of Materia Medica (SIMM), Chinese Academy of Sciences. We thank Q. Wang from SIMM for cryo-EM data collection. This work was supported by the National Science Foundation of China grants 31825010 (B.W.), 82121005 (B.W.), 32171439 (W.S.) and 31971362 (W.S.), the National Key R&D Program of China 2018YFA0507000 (B.W., Q.Z. and W.S.), the CAS Strategic Priority Research Program XDB37030100 (B.W. and Q.Z.) and the Shanghai Pilot Program for Basic Research—Chinese Academy of Sciences, Shanghai Branch JCYJ-SHFY-2021-008 (B.W.).

**Author contributions** X.Q. developed the protein expression and purification procedures, prepared the protein samples for cryo-EM studies, analysed the structural and functional data and wrote the first draft. N.Q. prepared the protein samples for cryo-EM and performed the functional assays. M.W. prepared cryo-samples, collected cryo-EM data, performed cryo-EM data processing and analysis, model building and structure refinement and helped with manuscript preparation. B.Z. identified lipid ligands by mass spectrometry. J.D. performed the functional assays and analysed the functional data. Z.Z. and W.X. helped with protein preparation and functional assays. X.C., L.M. and C.Y. expressed the proteins. S.H. helped with structure determination and data analysis. W.S. oversaw the lipid ligand identification study and edited the manuscript. Q.Z. and B.W. initiated the project, planned and analysed experiments, supervised the research and wrote the manuscript with input from all co-authors.

**Competing interests** The authors declare no competing interests.

**Additional information**
**Correspondence and requests for materials** should be addressed to Wenqing Shui, Qiang Zhao or Beili Wu.

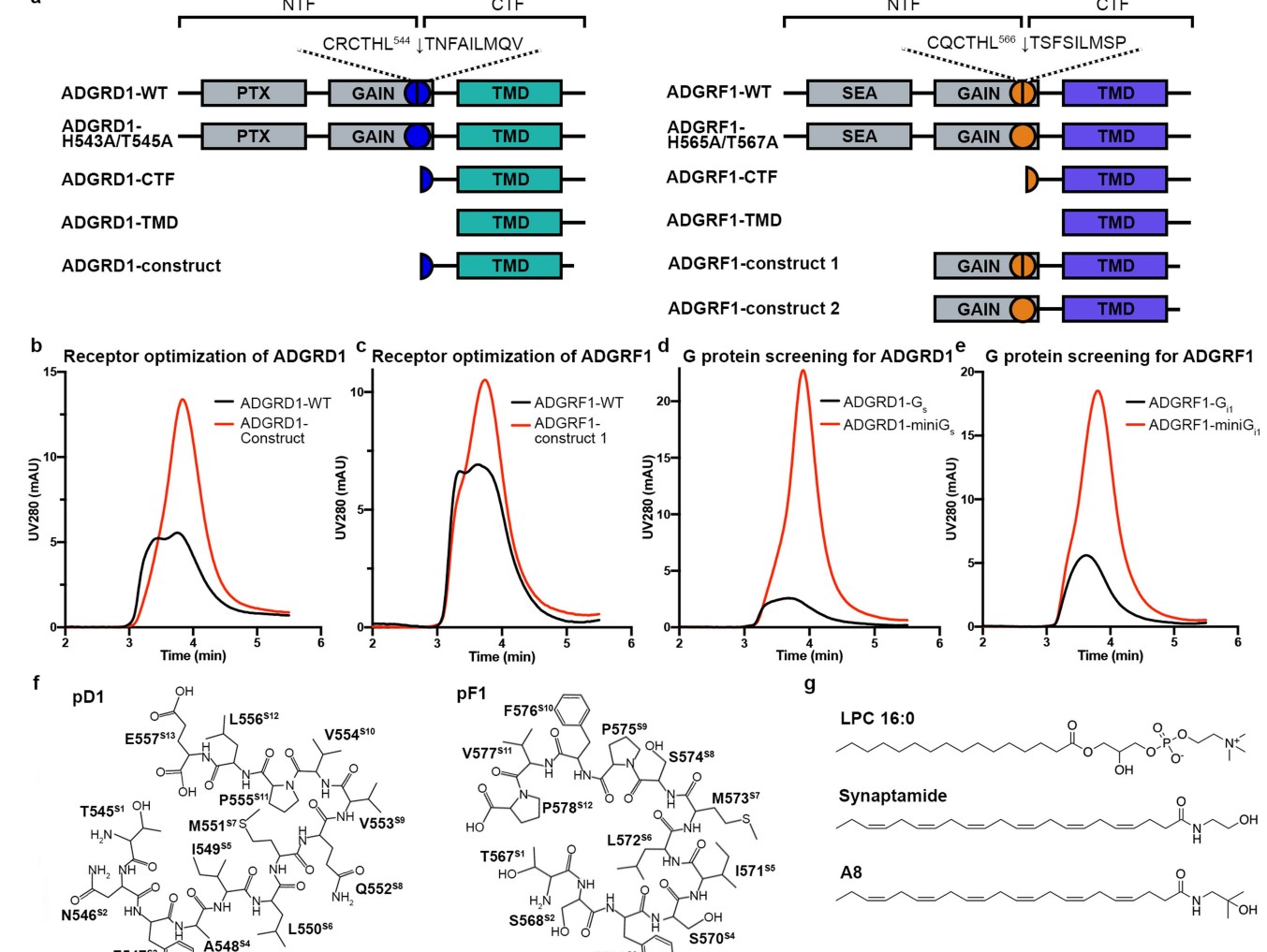

**Extended Data Fig. 1 | Protein optimization and ligand structures.**
**a**, Schematic diagrams of ADGRD1 and ADGRF1 constructs used in this study. ADGRD1-construct was used to determine the ADGRD1–miniG_s structure. ADGRF1-construct 1 was used to determine the ADGRF1–miniG_s and ADGRF1–miniG_{I1} structures. ADGRF1-construct 2 was used to determine the ADGRF1(H565A/T567A)–miniG_{I1} structure. PTX, pentraxin domain; SEA, sperm protein/enterokinase/agrin domain. **b**, **c**, Receptor optimization of ADGRD1 and ADGRF1 for the structural studies. The curves of analytical size-exclusion chromatography (aSEC) of purified protein samples show higher yield and better homogeneity for the optimized receptors. **d**, **e**, G protein screening for ADGRD1 and ADGRF1. The aSEC curves of purified receptor–G protein complexes show higher yield and better homogeneity for the miniG protein-bound receptors. **f**, Schematic diagrams of the stalk peptides pD1 and pF1. **g**, Chemical structures of LPC 16:0, synaptamide and A8.

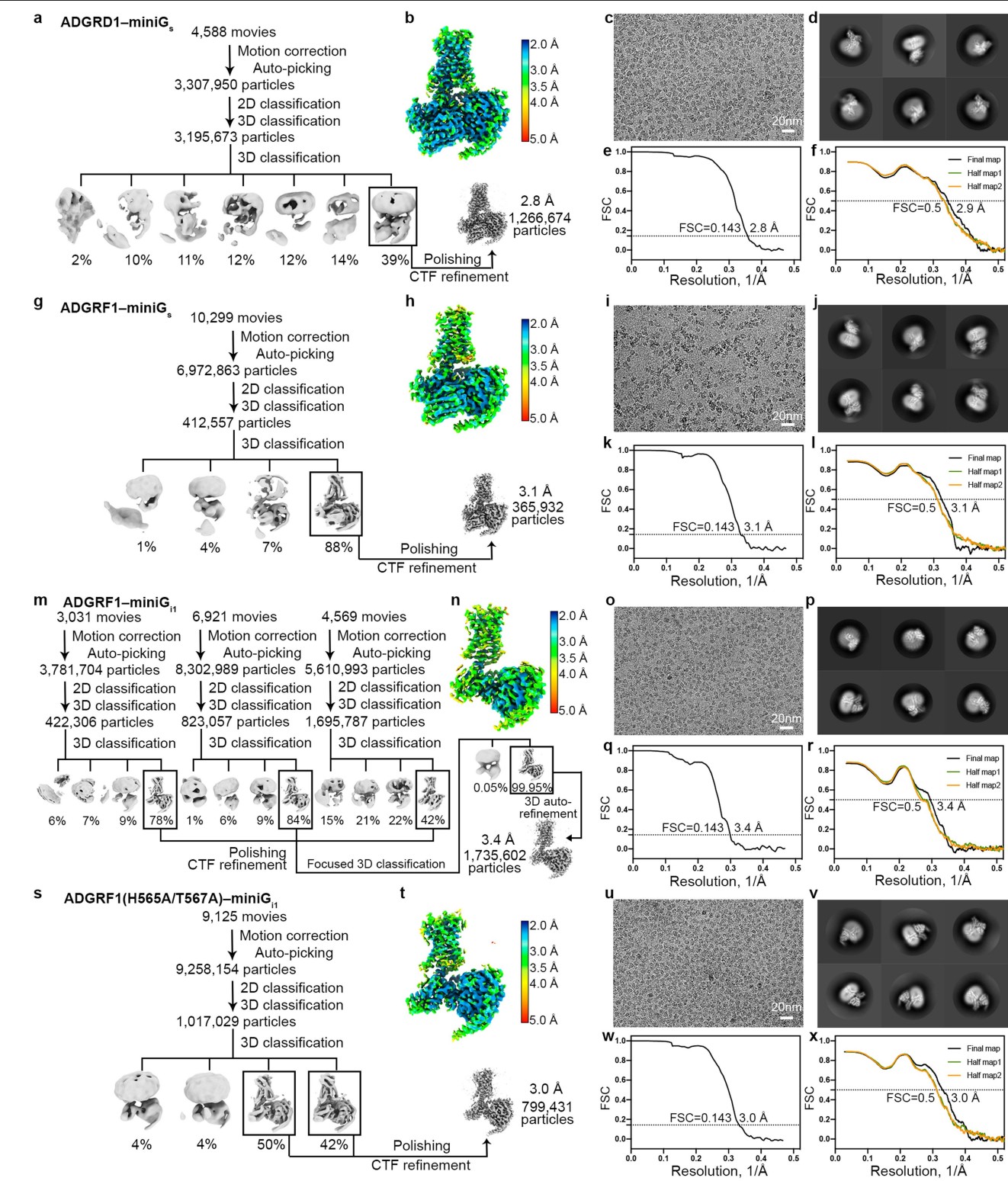

**Extended Data Fig. 2 | Cryo-EM processing and 3D reconstruction workflow.**
**a–f**, Results of ADGRD1–miniG$_s$. **a**, Data processing workflow. **b**, Cryo-EM map coloured according to local resolution (in Å). **c**, Representative cryo-EM image from one independent experiment. **d**, Two-dimensional (2D) averages. **e**, Gold-standard Fourier shell correlation (FSC) curve showing an overall resolution of 2.8 Å. **f**, Cross-validation of model to cryo-EM density map. FSC curves for the final model versus the final map and half maps are shown in black, green and yellow, respectively. **g–l**, Results of ADGRF1–miniG$_s$. **g**, Data processing workflow. **h**, Cryo-EM map coloured according to local resolution (in Å). **i**, Representative cryo-EM image from two independent experiments with similar results. **j**, 2D averages. **k**, Gold-standard FSC curve showing an overall resolution of 3.1 Å.

**l**, Cross-validation of model to cryo-EM density map. **m–r**, Results of ADGRF1–miniG$_{i1}$. **m**, Data processing workflow. **n**, Cryo-EM map coloured according to local resolution (in Å). **o**, Representative cryo-EM image from three independent experiments with similar results. **p**, 2D averages. **q**, Gold-standard FSC curve showing an overall resolution of 3.4 Å. **r**, Cross-validation of model to cryo-EM density map. **s–x**, Results of ADGRF1(H565A/T567A)–miniG$_{i1}$. **s**, Data processing workflow. **t**, Cryo-EM map coloured according to local resolution (in Å). **u**, Representative cryo-EM image from two independent experiments with similar results. **v**, 2D averages. **w**, Gold-standard FSC curve showing an overall resolution of 3.0 Å. **x**, Cross-validation of model to cryo-EM density map.

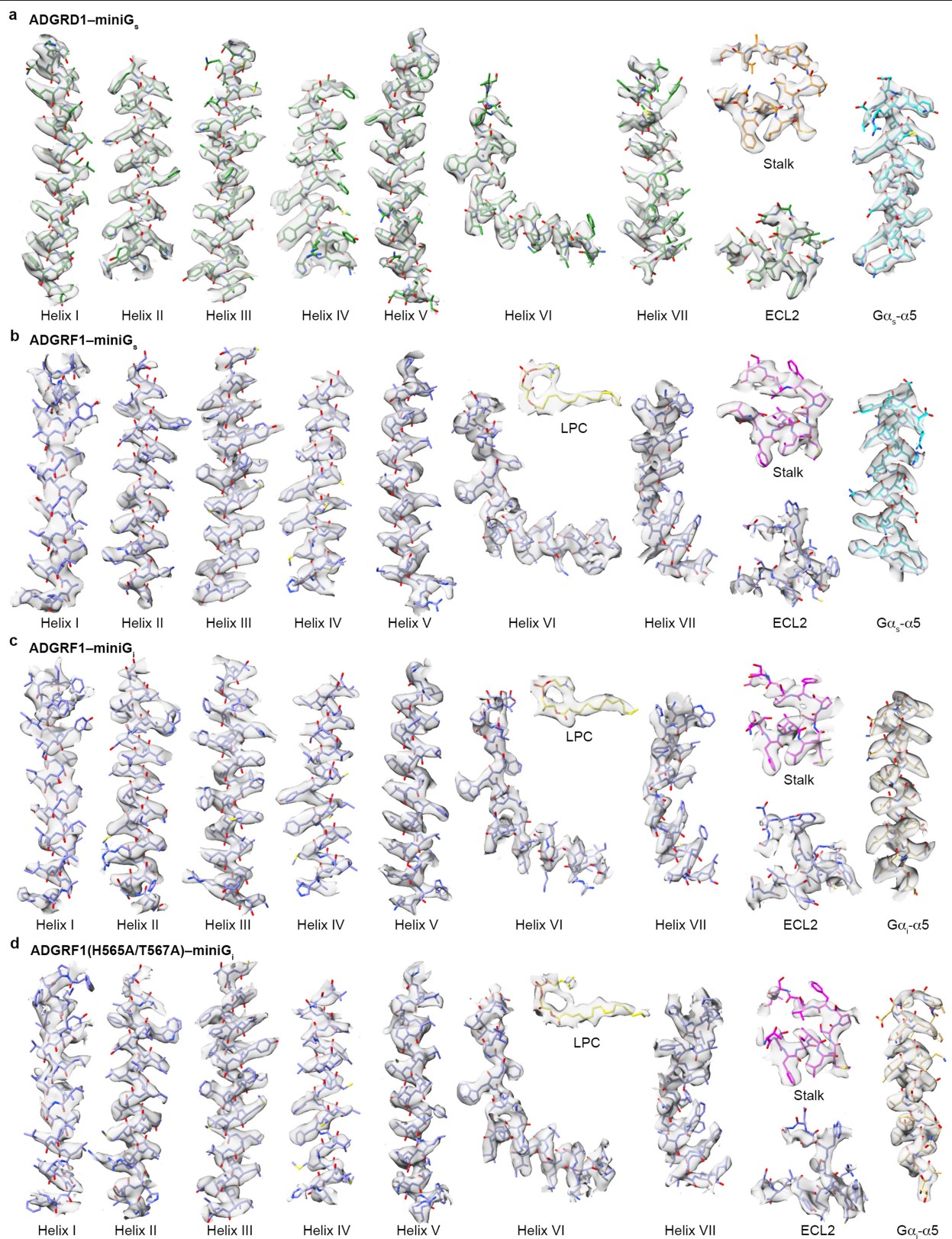

**Extended Data Fig. 3 | Cryo-EM density maps of the G protein-bound ADGRD1 and ADGRF1 structures. a**, ADGRD1–miniG$_s$; **b**, ADGRF1–miniG$_s$; **c**, ADGRF1–miniG$_{i1}$; **d**, ADGRF1(H565A/T567A)–miniG$_{i1}$. Cryo-EM maps and models of the four structures are shown for all transmembrane helices, stalk, ECL2, LPC and Gα α5-helix. The models are shown as sticks. The maps are coloured grey.

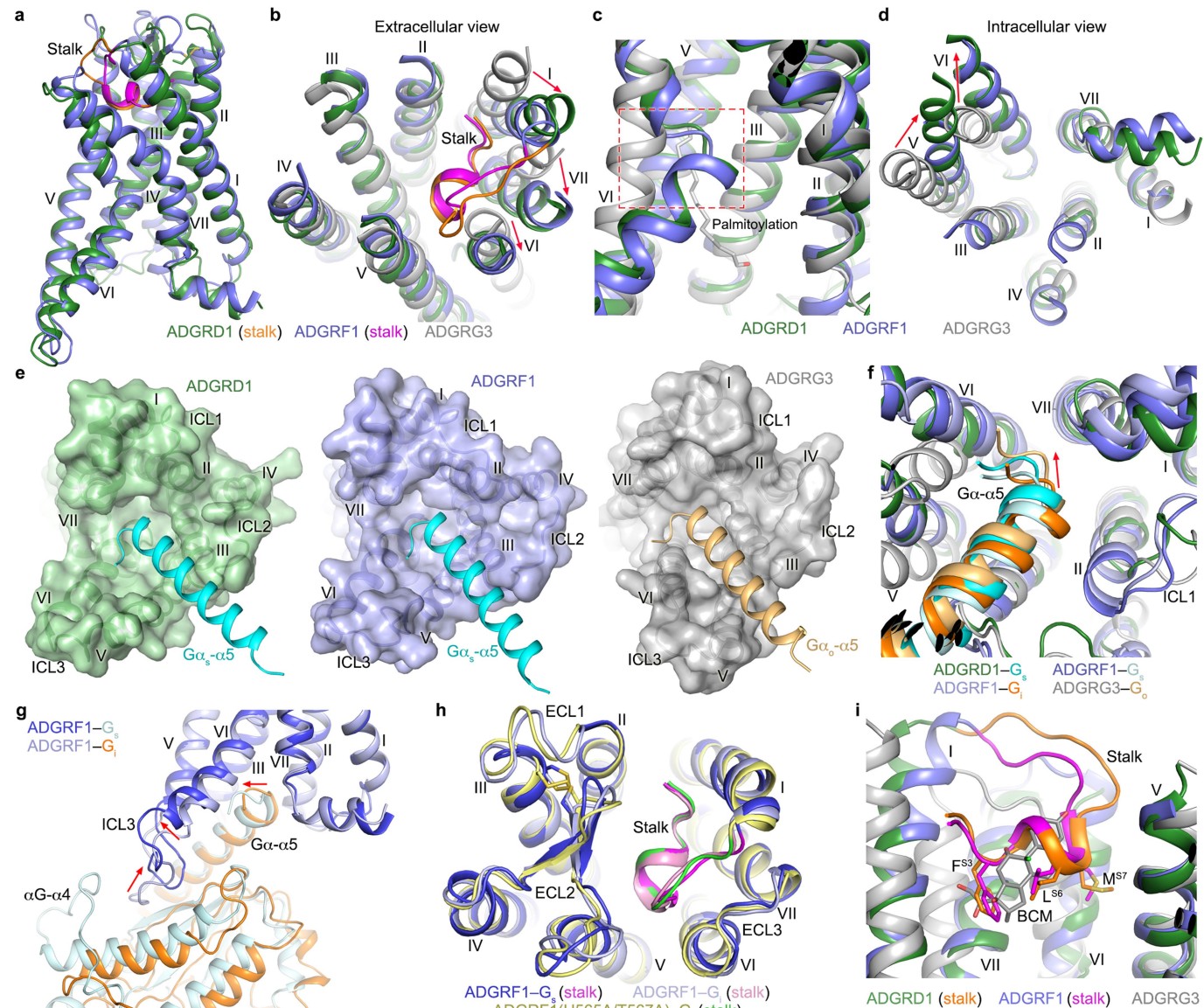

**Extended Data Fig. 4 | Comparison of aGPCR structures. a**, Structural comparison of the CTFs in ADGRD1 and ADGRF1. The receptors in the structures of ADGRD1–miniG$_s$ and ADGRF1–miniG$_s$ are shown in cartoon representation, and coloured green and blue, respectively. The stalks in the two receptors are coloured orange and magenta, respectively. **b–d**, Structural comparison of the helical bundles in ADGRD1, ADGRF1 and ADGRG3. The transmembrane helical bundles in the structures of ADGRD1–miniG$_s$ and ADGRF1–miniG$_s$ and the beclomethasone (BCM)–ADGRG3–G$_o$ structure (PDB ID: 7D76) are shown in cartoon representation. **b**, Extracellular view. The red arrows indicate the movements of helices I, VI and VII in ADGRD1 and ADGRF1 relative to those in ADGRG3. **c**, Comparison of helix VI conformation. The sharp kink of helix VI in ADGRD1 and ADGRF1 is highlighted by a red dashed box. The palmitoylation in the ADGRG3 structure is shown as grey sticks. **d**, Intracellular view. The red arrows indicate the movements of helices V and VI in ADGRD1 and ADGRF1 relative to those in ADGRG3. **e**, Comparison of the G protein-binding cavities in ADGRD1, ADGRF1 and ADGRG3. The receptors in the structures of ADGRD1–miniG$_s$, ADGRF1–miniG$_s$ and BCM–ADGRG3–G$_o$ are shown in cartoon

and surface representations. The α5-helix in Gα is coloured cyan (Gα$_s$) and gold (Gα$_o$). **f**, Comparison of the Gα α5-helix binding poses in ADGRD1, ADGRF1 and ADGRG3. The structures of ADGRD1–miniG$_s$, ADGRF1–miniG$_s$, ADGRF1–miniG$_{i1}$ and BCM–ADGRG3–G$_o$ are shown in an intracellular view. The red arrow indicates the movement of the C terminus of Gα α5-helix in the ADGRG3 structure relative to that in the ADGRD1 and ADGRF1 structures. **g**, Comparison of G$_s$ and G$_i$ binding in ADGRF1. The ADGRF1–miniG$_s$ and ADGRF1–miniG$_{i1}$ structures are shown in cartoon representation. The red arrows indicate the movements of the intracellular tip of helix VI, ICL3 and the C terminus of Gα α5-helix in the ADGRF1–miniG$_s$ structure relative to those in the ADGRF1–miniG$_i$ structure. **h**, Comparison of the stalk conformation in the ADGRF1 structures. The structures of ADGRF1–miniG$_s$, ADGRF1–miniG$_{i1}$ and ADGRF1(H565A/T567A)–miniG$_{i1}$ are shown in an extracellular view. **i**, Comparison of the binding sites for the stalk in ADGRD1 and ADGRF1 and the ligand glucocorticoid in ADGRG3. The structures of ADGRD1–miniG$_s$, ADGRF1–miniG$_s$ and BCM–ADGRG3–G$_o$ are shown. The stalk residues F$^{S3}$, L$^{S6}$ and M$^{S7}$ are shown as sticks. The glucocorticoid BCM in the ADGRG3 structure is shown as grey sticks.

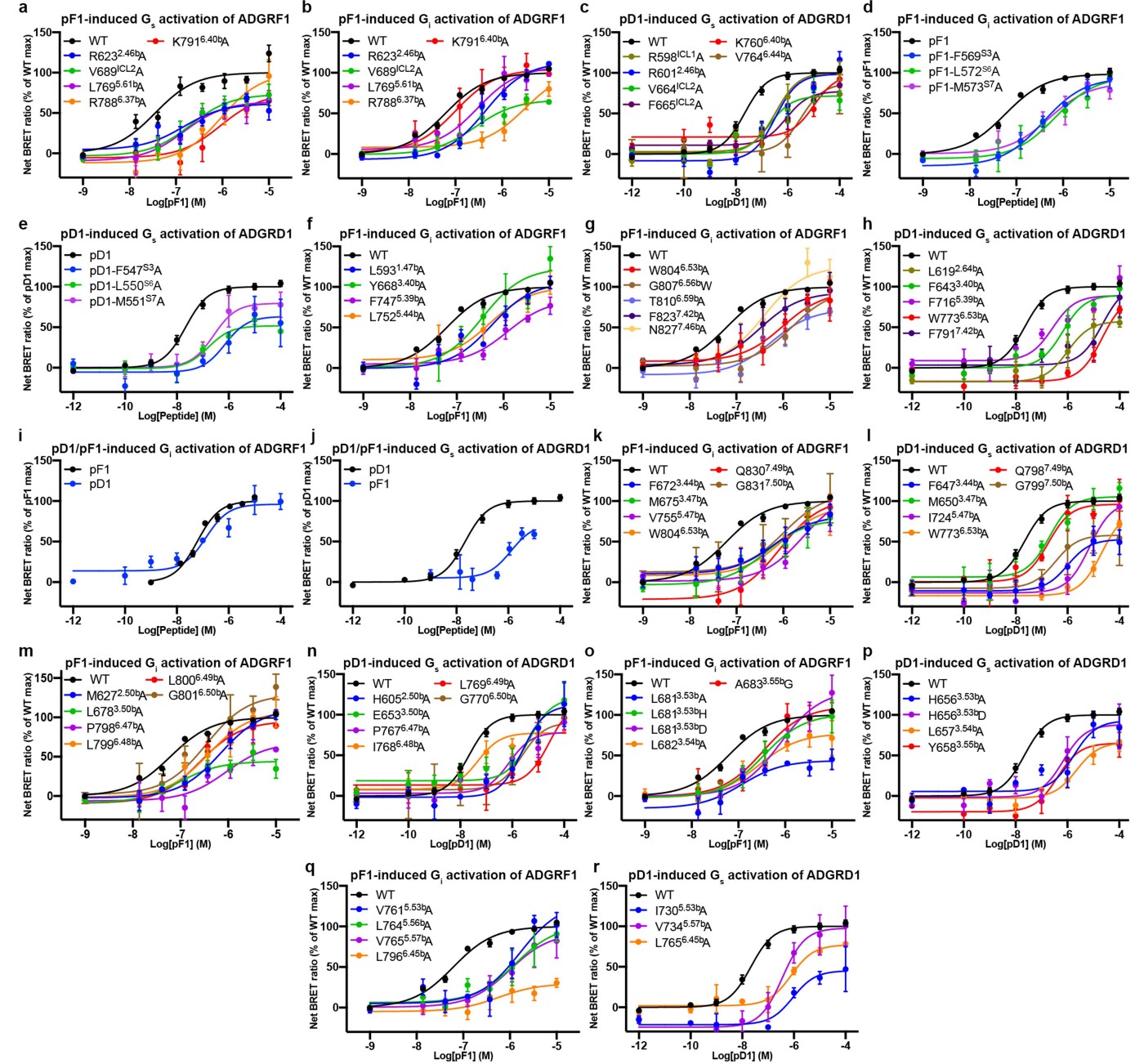

**Extended Data Fig. 5 | Synthetic stalk peptide-induced G protein activation of wild-type ADGRD1 and ADGRF1 and mutants using BRET assays.** Data are shown as mean ± s.e.m. from at least three independent experiments performed in technical duplicate. Extended Data Table 3 provides detailed numbers of independent experiments (*n*), statistical evaluation and expression level.

**Extended Data Fig. 6 | Sequence alignment of aGPCRs.** The stalk-N is highlighted by a red background. Some key positions in the TMD are highlighted by a green background. The alignment was generated using UniProt (http://www.uniprot.org/align/) and the graphic was prepared on the ESPript 3.0 server (http://espript.ibcp.fr/ESPript/cgi-bin/ESPript.cgi).

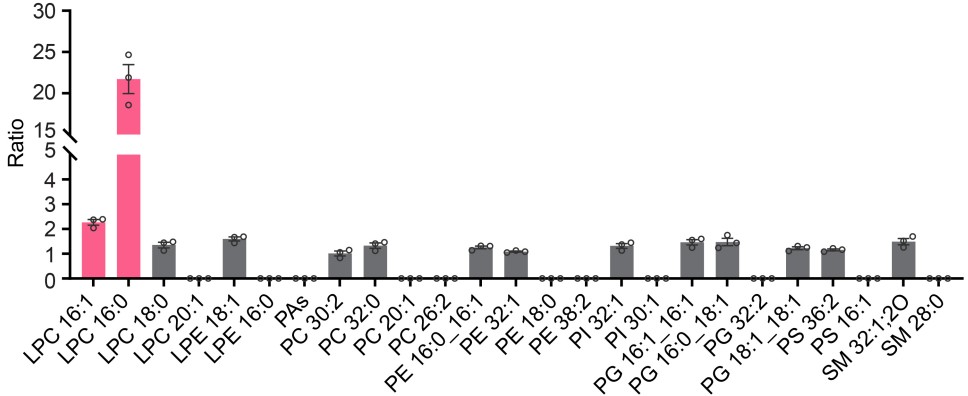

**Extended Data Fig. 7 | EIC peak ratios of identified phospholipids associated with ADGRF1 versus ADGRD1.** Representative phospholipids in different classes are shown, with their EIC peak ratios indicating the compound abundance in ADGRF1 versus ADGRD1. Two specific binders to ADGRF1, LPC 16:0 and LPC 16:1, were distinguished with a mean ratio > 2 and $P$ < 0.05, and are highlighted in pink. Data are presented as mean ± s.e.m. (bars) from three independent experiments performed in technical triplicate with individual data points shown (dots).

**Extended Data Table 1 | Cryo-EM data collection, refinement and validation statistics**

| | ADGRD1– miniG$_s$ (EMDB-32817) (PDB 7WU2) | ADGRF1– miniG$_s$ (EMDB-32818) (PDB 7WU3) | ADGRF1– miniG$_{i1}$ (EMDB-32819) (PDB 7WU4) | ADGRF1(H565A/ T567A)–miniG$_{i1}$ (EMDB-32820) (PDB 7WU5) |
|---|---|---|---|---|
| **Data collection and processing** | | | | |
| Magnification | 81,000 | 81,000 | 81,000 | 81,000 |
| Voltage (kV) | 300 | 300 | 300 | 300 |
| Electron exposure (e$^-$/Å$^2$) | 70 | 70 | 70 | 70 |
| Defocus range ($\mu$m) | −0.8 ~ −1.5 | −0.8 ~ −1.5 | −0.8 ~ −1.5 | −0.8 ~ −1.5 |
| Pixel size (Å) | 1.071 | 1.071 | 1.071 | 1.071 |
| Symmetry imposed | C1 | C1 | C1 | C1 |
| Initial particle images (no.) | 3,307,950 | 6,972,863 | 17,695,686 | 9,258,154 |
| Final particle images (no.) | 1,266,674 | 365,932 | 1,735,602 | 799,431 |
| Map resolution (Å) | 2.8 | 3.1 | 3.4 | 3.0 |
| FSC threshold | 0.143 | 0.143 | 0.143 | 0.143 |
| Map resolution range (Å) | 2.0-5.0 | 2.0-5.0 | 2.0-5.0 | 2.0-5.0 |
| | | | | |
| **Refinement** | | | | |
| Initial model used (PDB code) | 6LMK | 6LMK | 6LML | 6LML |
| Model resolution (Å) | 2.9 | 3.1 | 3.4 | 3.0 |
| FSC threshold | 0.5 | 0.5 | 0.5 | 0.5 |
| Map sharpening $B$ factor (Å$^2$) | −91 | −78 | −134 | −90 |
| Model composition | | | | |
| Non-hydrogen atoms | 7,829 | 7,877 | 6,887 | 6,948 |
| Protein residues | 1,010 | 1,013 | 888 | 888 |
| Lipid atoms | 0 | 33 | 61 | 61 |
| $B$ factors (Å$^2$) | | | | |
| Protein | 45.47 | 68.08 | 58.27 | 81.19 |
| Lipid | - | 83.07 | 85.15 | 120.45 |
| R.m.s. deviations | | | | |
| Bond lengths (Å) | 0.002 | 0.003 | 0.002 | 0.003 |
| Bond angles (°) | 0.513 | 0.523 | 0.480 | 0.501 |
| Validation | | | | |
| Molprobity score | 1.38 | 1.49 | 1.51 | 1.56 |
| Clashscore | 6.44 | 7.30 | 7.40 | 8.96 |
| Poor rotamers (%) | 0.00 | 0.00 | 0.00 | 0.00 |
| Ramachandran plot | | | | |
| Favored (%) | 97.90 | 97.60 | 97.49 | 97.61 |
| Allowed (%) | 2.10 | 2.40 | 2.51 | 2.39 |
| Disallowed (%) | 0.00 | 0.00 | 0.00 | 0.00 |

**Extended Data Table 2 | Basal activity of wild-type ADGRD1 and ADGRF1 and mutants, measured by cAMP and inositol phosphate accumulation assays**

| | | ADGRF1 | | | | | | | | ADGRD1 | | | | | |
|---|---|---|---|---|---|---|---|---|---|---|---|---|---|---|---|
| | | cAMP accumulation | | | IP accumulation | | | Expression | | | cAMP accumulation | | | Expression | |
| | Mutants[†] | % of WT[‡] | P value | n[§] | % of WT[‡] | P value | n[§] | % of WT | P value | Mutants[†] | % of WT[‡] | P value | n[§] | % of WT | P value |
| | WT | 100 | / | 52 | 100 | / | 52 | 100 | / | WT | 100 | / | 32 | 100 | / |
| | H565A/T567A | 99 ± 10 | 0.9999 | 6 | 95 ± 3 | 0.9987 | 6 | 92 ± 2 | 0.9985 | H543A/T545A | 109 ± 8 | 0.4241 | 7 | 99 ± 6 | >0.9999 |
| Stalk | $T567^{S1}A$ | 71 ± 5** | 0.0004 | 5 | 75 ± 3*** | <0.0001 | 6 | 83 ± 4 | 0.2749 | $T545^{S1}A$ | 8 ± 3*** | <0.0001 | 5 | 113 ± 31 | 0.9988 |
| | $S568^{S2}A$ | 49 ± 7*** | <0.0001 | 5 | 52 ± 2*** | <0.0001 | 5 | 90 ± 2 | 0.9834 | $N546^{S2}A$ | 17 ± 5*** | <0.0001 | 5 | 92 ± 6 | 0.9994 |
| | $F569^{S3}A$ | -5 ± 3*** | <0.0001 | 6 | -2 ± 1*** | <0.0001 | 8 | 88 ± 4 | 0.9218 | $F547^{S3}A$ | 8 ± 1*** | <0.0001 | 5 | 82 ± 1 | 0.9839 |
| | $S570^{S4}A$ | 42 ± 5*** | <0.0001 | 5 | 16 ± 1*** | <0.0001 | 6 | 94 ± 7 | 0.9990 | $A548^{S4}G$ | 100 ± 6 | >0.9999 | 5 | 82 ± 8 | 0.9854 |
| | $I571^{S5}A$ | 40 ± 5*** | <0.0001 | 6 | 53 ± 3*** | <0.0001 | 6 | 92 ± 4 | 0.9987 | $I549^{S5}A$ | 11 ± 1*** | <0.0001 | 5 | 52 ± 3* | 0.0019 |
| | $L572^{S6}A$ | 18 ± 3*** | <0.0001 | 5 | 0 ± 1*** | <0.0001 | 7 | 101 ± 6 | >0.9999 | $L550^{S6}A$ | 0 ± 1*** | <0.0001 | 5 | 41 ± 2*** | <0.0001 |
| | $M573^{S7}A$ | 19 ± 7*** | <0.0001 | 5 | 3 ± 2*** | <0.0001 | 6 | 97 ± 6 | 0.9996 | $M551^{S7}A$ | 9 ± 2*** | <0.0001 | 5 | 50 ± 2* | 0.0012 |
| | $S574^{S8}A$ | 50 ± 2*** | <0.0001 | 5 | 67 ± 1*** | <0.0001 | 6 | 97 ± 6 | 0.9997 | $Q552^{S8}A$ | 10 ± 3*** | <0.0001 | 5 | 11 ± 3*** | <0.0001 |
| | $P575^{S9}A$ | 44 ± 8*** | <0.0001 | 5 | 55 ± 5*** | <0.0001 | 6 | 101 ± 5 | 0.9998 | $V553^{S9}A$ | 29 ± 6*** | <0.0001 | 5 | 39 ± 2*** | <0.0001 |
| | $F576^{S10}A$ | 48 ± 3*** | <0.0001 | 5 | 59 ± 2*** | <0.0001 | 7 | 102 ± 3 | 0.9998 | $V554^{S10}A$ | 19 ± 5*** | <0.0001 | 5 | 97 ± 17 | 0.9998 |
| | $V577^{S11}A$ | 25 ± 3*** | <0.0001 | 5 | 39 ± 2*** | <0.0001 | 7 | 92 ± 8 | 0.9985 | $P555^{S11}A$ | 38 ± 2*** | <0.0001 | 5 | 39 ± 3*** | <0.0001 |
| | $P578^{S12}A$ | 37 ± 4*** | <0.0001 | 7 | 28 ± 2*** | <0.0001 | 7 | 97 ± 5 | 0.9996 | | | | | | |
| Stalk binding pocket | $V585^{1.39b}A$ | 30 ± 5*** | <0.0001 | 5 | 66 ± 7*** | <0.0001 | 5 | 99 ± 7 | >0.9999 | $Q563^{1.36b}A$ | 10 ± 2*** | <0.0001 | 5 | 90 ± 6 | 0.9991 |
| | $T589^{1.43b}A$ | 32 ± 5*** | <0.0001 | 5 | 4 ± 1*** | <0.0001 | 5 | 80 ± 1 | 0.0637 | $L566^{1.39b}A$ | 2 ± 1*** | <0.0001 | 5 | 83 ± 7 | 0.9980 |
| | $L593^{1.47b}A$ | 21 ± 5*** | <0.0001 | 5 | 62 ± 3*** | <0.0001 | 7 | 85 ± 6 | 0.4468 | $S570^{1.43b}A$ | 23 ± 4*** | <0.0001 | 5 | 126 ± 21 | 0.5913 |
| | $F641^{2.64b}A$ | 43 ± 6*** | <0.0001 | 5 | 10 ± 2*** | <0.0001 | 5 | 19 ± 2*** | <0.0001 | $L619^{2.64b}A$ | 38 ± 6*** | <0.0001 | 6 | 95 ± 7 | 0.9997 |
| | $Y668^{3.40b}A$ | 30 ± 4*** | <0.0001 | 5 | 3 ± 2*** | <0.0001 | 6 | 79 ± 6* | 0.0225 | $F643^{3.40b}A$ | 60 ± 5*** | <0.0001 | 5 | 82 ± 6 | 0.9854 |
| | $V732^{ECL2}A$ | 89 ± 7 | 0.9373 | 7 | 88 ± 8 | 0.1671 | 5 | 89 ± 5 | 0.9821 | $N703^{ECL2}A$ | 53 ± 4*** | <0.0001 | 5 | 43 ± 1*** | <0.0001 |
| | $W734^{ECL2}A$ | 9 ± 5*** | <0.0001 | 5 | 19 ± 3*** | <0.0001 | 5 | 71 ± 2*** | <0.0001 | $W705^{ECL2}A$ | 1 ± 1*** | <0.0001 | 5 | 49 ± 11** | 0.0007 |
| | $F747^{5.39b}A$ | 8 ± 4*** | <0.0001 | 5 | 1 ± 1*** | <0.0001 | 5 | 94 ± 2 | 0.9992 | $F716^{5.39b}A$ | 33 ± 7*** | <0.0001 | 5 | 90 ± 11 | 0.9992 |
| | $V748^{5.40b}A$ | 73 ± 8* | 0.0023 | 5 | 73 ± 3*** | <0.0001 | 5 | 106 ± 5 | 0.9992 | $V717^{5.40b}A$ | 45 ± 7*** | <0.0001 | 6 | 87 ± 6 | 0.9987 |
| | $L752^{5.44b}A$ | 13 ± 6*** | <0.0001 | 5 | 31 ± 2*** | <0.0001 | 6 | 102 ± 4 | 0.9997 | | | | | | |
| | $W804^{6.53b}A$ | 36 ± 7*** | <0.0001 | 5 | 7 ± 4*** | <0.0001 | 5 | 89 ± 4 | 0.9787 | $W773^{6.53b}A$ | 4 ± 1*** | <0.0001 | 5 | 54 ± 14* | 0.0040 |
| | $G807^{6.56b}W$ | 49 ± 3*** | <0.0001 | 5 | 11 ± 2*** | <0.0001 | 5 | 94 ± 5 | 0.9991 | | | | | | |
| | $T810^{6.59b}A$ | 29 ± 4*** | <0.0001 | 5 | 48 ± 1*** | <0.0001 | 5 | 81 ± 9 | 0.0725 | | | | | | |
| | $H820^{7.39b}A$ | 43 ± 3*** | <0.0001 | 5 | 7 ± 2*** | <0.0001 | 6 | 97 ± 9 | 0.9996 | $Q788^{7.39b}A$ | 29 ± 4*** | <0.0001 | 5 | 27 ± 1*** | <0.0001 |
| | $F823^{7.42b}A$ | 33 ± 2*** | <0.0001 | 5 | 6 ± 3*** | <0.0001 | 5 | 28 ± 16*** | <0.0001 | $F791^{7.42b}A$ | 13 ± 2*** | <0.0001 | 5 | 40 ± 4*** | <0.0001 |
| | $N827^{7.46b}A$ | 49 ± 6*** | <0.0001 | 5 | 15 ± 5*** | <0.0001 | 5 | 34 ± 4*** | <0.0001 | $N795^{7.46b}A$ | 10 ± 3*** | <0.0001 | 5 | 37 ± 2*** | <0.0001 |
| Signaling cascade | $F672^{3.44b}A$ | 25 ± 2*** | <0.0001 | 7 | 13 ± 2*** | <0.0001 | 7 | 91 ± 6 | 0.9981 | $F647^{3.44b}A$ | 4 ± 0*** | <0.0001 | 5 | 10 ± 2*** | <0.0001 |
| | $M675^{3.47b}A$ | 12 ± 2*** | <0.0001 | 7 | 13 ± 2*** | <0.0001 | 7 | 101 ± 5 | 0.9999 | $M650^{3.47b}A$ | 5 ± 1*** | <0.0001 | 5 | 19 ± 3*** | <0.0001 |
| | $V755^{5.47b}A$ | 49 ± 4*** | <0.0001 | 5 | 65 ± 6*** | <0.0001 | 5 | 158 ± 2*** | <0.0001 | $I724^{5.47b}A$ | 33 ± 5*** | <0.0001 | 5 | 66 ± 2 | 0.1356 |
| | $W804^{6.53b}A$ | 36 ± 7*** | <0.0001 | 5 | 7 ± 4*** | <0.0001 | 5 | 89 ± 4 | 0.9787 | $W773^{6.53b}A$ | 4 ± 1*** | <0.0001 | 5 | 54 ± 14* | 0.0040 |
| | $Q830^{7.49b}A$ | 37 ± 7*** | <0.0001 | 5 | 36 ± 3*** | <0.0001 | 5 | 109 ± 9 | 0.9982 | $Q797^{7.49b}A$ | 7 ± 3*** | <0.0001 | 5 | 102 ± 10 | 0.9998 |
| | $G831^{7.50b}A$ | 37 ± 6*** | <0.0001 | 5 | 30 ± 3*** | <0.0001 | 5 | 75 ± 15* | 0.0021 | $G799^{7.50b}A$ | 3 ± 2*** | <0.0001 | 5 | 80 ± 6 | 0.9632 |
| | $M627^{2.50b}A$ | 44 ± 7*** | <0.0001 | 5 | 27 ± 5*** | <0.0001 | 5 | 79 ± 2* | 0.0235 | $H605^{2.50b}A$ | 5 ± 1*** | <0.0001 | 5 | 86 ± 13 | 0.9986 |
| | $L678^{3.50b}A$ | 48 ± 4*** | <0.0001 | 5 | 104 ± 7 | 0.9988 | 6 | 59 ± 8*** | <0.0001 | $E653^{3.50b}A$ | 8 ± 3*** | <0.0001 | 5 | 82 ± 9 | 0.9839 |
| | $P798^{6.47b}A$ | 44 ± 4*** | <0.0001 | 5 | 8 ± 2*** | <0.0001 | 5 | 37 ± 8*** | <0.0001 | $P767^{6.47b}A$ | 5 ± 1*** | <0.0001 | 5 | 84 ± 2 | 0.9983 |
| | $L799^{6.48b}A$ | 31 ± 7*** | <0.0001 | 5 | 41 ± 6*** | <0.0001 | 5 | 82 ± 2 | 0.1124 | $I768^{6.48b}A$ | 5 ± 2*** | <0.0001 | 5 | 76 ± 4 | 0.7907 |
| | $L800^{6.49b}A$ | 35 ± 6*** | <0.0001 | 5 | -1 ± 1*** | <0.0001 | 5 | 120 ± 10* | 0.0477 | $L769^{6.49b}A$ | 20 ± 7*** | <0.0001 | 5 | 71 ± 30 | 0.3775 |
| | $G801^{6.50b}A$ | 23 ± 6*** | <0.0001 | 6 | 2 ± 2*** | <0.0001 | 6 | 115 ± 2 | 0.5317 | $G770^{6.50b}A$ | 18 ± 6*** | <0.0001 | 5 | 133 ± 45 | 0.1885 |
| | $L681^{3.53b}A$ | 215 ± 17*** | <0.0001 | 5 | 82 ± 3** | 0.0005 | 5 | 110 ± 12 | 0.9852 | $H656^{3.53b}A$ | 3 ± 1*** | <0.0001 | 5 | 104 ± 13 | 0.9997 |
| | $L681^{3.53b}H$ | 106 ± 20 | 0.9991 | 6 | 108 ± 7 | 0.7304 | 6 | 85 ± 10 | 0.5757 | | | | | | |
| | $L681^{3.53b}D$ | 57 ± 5*** | <0.0001 | 5 | 62 ± 4*** | <0.0001 | 5 | 90 ± 7 | 0.9851 | $H656^{3.53b}D$ | 3 ± 1*** | <0.0001 | 5 | 56 ± 2* | 0.0075 |
| | $L682^{3.54b}A$ | 45 ± 7*** | <0.0001 | 5 | 16 ± 2*** | <0.0001 | 5 | 93 ± 8 | 0.9987 | $L657^{3.54b}A$ | 2 ± 1*** | <0.0001 | 5 | 7 ± 2*** | <0.0001 |
| | $A683^{3.55b}G$ | 51 ± 7*** | <0.0001 | 5 | 68 ± 7*** | <0.0001 | 5 | 99 ± 3 | 0.9999 | $Y658^{3.55b}A$ | 13 ± 2*** | <0.0001 | 5 | 91 ± 3 | 0.9993 |
| | $V761^{5.53b}A$ | 27 ± 5*** | <0.0001 | 5 | 22 ± 4*** | <0.0001 | 5 | 81 ± 8 | 0.0968 | $I730^{5.53b}A$ | 8 ± 4*** | <0.0001 | 5 | 37 ± 8*** | <0.0001 |
| | $L764^{5.56b}A$ | 29 ± 5*** | <0.0001 | 5 | 26 ± 6*** | <0.0001 | 5 | 79 ± 6* | 0.0216 | | | | | | |
| | $V765^{5.57b}A$ | 53 ± 7*** | <0.0001 | 5 | 46 ± 6*** | <0.0001 | 5 | 82 ± 9 | 0.1324 | $V734^{5.57b}A$ | 11 ± 3*** | <0.0001 | 5 | 97 ± 9 | 0.9997 |
| | $L796^{6.45b}A$ | 43 ± 5*** | <0.0001 | 5 | 25 ± 2*** | <0.0001 | 5 | 74 ± 12* | 0.0010 | $L765^{6.45b}A$ | 8 ± 2*** | <0.0001 | 5 | 58 ± 7* | 0.0138 |
| Lipid binding pocket | $I625^{2.48b}A$ | 77 ± 8* | 0.0213 | 5 | 73 ± 3*** | <0.0001 | 5 | 36 ± 4*** | <0.0001 | | | | | | |
| | $W674^{3.46b}A$ | 16 ± 3*** | <0.0001 | 5 | 1 ± 1*** | <0.0001 | 5 | 11 ± 2*** | <0.0001 | | | | | | |
| | $M677^{3.49b}A$ | 37 ± 8*** | <0.0001 | 5 | 91 ± 4 | 0.6987 | 5 | 91 ± 3 | 0.9908 | | | | | | |
| | $I680^{3.52b}A$ | 30 ± 6*** | <0.0001 | 5 | 83 ± 5** | 0.0006 | 5 | 88 ± 7 | 0.9037 | | | | | | |
| | $Y684^{3.56b}A$ | 17 ± 6*** | <0.0001 | 5 | 36 ± 1*** | <0.0001 | 6 | 103 ± 3 | 0.9996 | | | | | | |
| | $L688^{ICL2}A$ | 37 ± 6*** | <0.0001 | 5 | 74 ± 6*** | <0.0001 | 6 | 112 ± 8 | 0.9501 | | | | | | |
| | $H691^{ICL2}A$ | 23 ± 3*** | <0.0001 | 5 | 66 ± 4*** | <0.0001 | 5 | 90 ± 8 | 0.9857 | | | | | | |
| | $M693^{ICL2}A$ | 39 ± 6*** | <0.0001 | 5 | 80 ± 2*** | <0.0001 | 5 | 98 ± 3 | 0.9997 | | | | | | |
| | $M698^{4.41b}A$ | 70 ± 8** | 0.0002 | 5 | 82 ± 7** | 0.0005 | 5 | 90 ± 7 | 0.9849 | | | | | | |
| | $M699^{4.42b}A$ | 39 ± 5*** | <0.0001 | 5 | 71 ± 2*** | <0.0001 | 5 | 78 ± 5* | 0.0177 | | | | | | |
| | $V701^{4.44b}A$ | 73 ± 8* | 0.0016 | 5 | 82 ± 2** | 0.0006 | 5 | 92 ± 5 | 0.9984 | | | | | | |
| | $G702^{4.45b}Y$ | 39 ± 6*** | <0.0001 | 5 | 13 ± 4*** | <0.0001 | 5 | 86 ± 3 | 0.6118 | | | | | | |
| | $F703^{4.46b}A$ | 20 ± 3*** | <0.0001 | 5 | 67 ± 4*** | <0.0001 | 5 | 75 ± 8* | 0.0015 | | | | | | |
| G protein binding pocket | $R623^{2.46b}A$ | 37 ± 4*** | <0.0001 | 5 | 4 ± 1*** | <0.0001 | 5 | 96 ± 6 | 0.9995 | $R598^{ICL1}A$ | 64 ± 3*** | <0.0001 | 5 | 99 ± 4 | >0.9999 |
| | $V689^{ICL2}A$ | 78 ± 8* | 0.0378 | 5 | 36 ± 6*** | <0.0001 | 5 | 88 ± 5 | 0.9223 | $R601^{2.46b}A$ | 15 ± 4*** | <0.0001 | 5 | 83 ± 6 | 0.9980 |
| | $F690^{ICL2}A$ | 48 ± 5*** | <0.0001 | 5 | 13 ± 4*** | <0.0001 | 5 | 84 ± 3 | 0.4033 | $V664^{ICL2}A$ | 33 ± 6*** | <0.0001 | 5 | 93 ± 2 | 0.9995 |
| | $L769^{5.61b}A$ | 54 ± 8*** | <0.0001 | 7 | 59 ± 5*** | <0.0001 | 7 | 99 ± 6 | 0.9999 | $F665^{ICL2}A$ | 9 ± 3*** | <0.0001 | 5 | 107 ± 8 | 0.9995 |
| | $R788^{6.37b}A$ | 15 ± 6*** | <0.0001 | 5 | 96 ± 8 | 0.9991 | 6 | 127 ± 3** | 0.0007 | $K760^{6.40b}A$ | 13 ± 1*** | <0.0001 | 5 | 104 ± 23 | 0.9997 |
| | $K791^{6.40b}A$ | 51 ± 7*** | <0.0001 | 6 | 11 ± 2*** | <0.0001 | 5 | 101 ± 11 | 0.9998 | $V764^{6.44b}A$ | 14 ± 4*** | <0.0001 | 5 | 94 ± 10 | 0.9996 |

[†]All mutations were introduced in the wild-type receptor.

[‡]Data are mean ± s.e.m. from at least five independent experiments. *$P<0.05$, **$P<0.001$, ***$P<0.0001$ by one-way analysis of variance followed by Dunnett's post-test compared to the response of wild type.

[§]Sample size, the number of independent experiments performed in technical triplicate.

[‖]Protein expression levels of ADGRD1 and ADGRF1 constructs at the cell surface were determined in parallel by flow cytometry with an anti-Flag antibody (Sigma) and reported as per cent compared to the wild type from three independent measurements performed in technical duplicate. The mutants with low expression level (less than 40% of wild-type expression level) are indicated with a grey background.

## Extended Data Table 3 | G protein activation of wild-type ADGRD1 and ADGRF1 and mutants, measured by BRET assay

### pF1-induced $G_i$ activation of ADGRF1

| Mutants[†] | $EC_{50}$ (nM) | $EC_{50}$ ratio[‡] | pEC50 mean ± s.e.m.[§] | P value | $E_{max}$[‖] % of WT[§] | P value | n[¶] | Expression[#] % of WT | P value |
|---|---|---|---|---|---|---|---|---|---|
| WT | 61 | 1 | 7.21 ± 0.06 | / | 100 ± 3 | / | 30 | 100 | / |
| WT-pD1 | 133 | 2 | 6.88 ± 0.24 | 0.9986 | 82 ± 8 | 0.9984 | 3 | 100 | / |
| Construct 1[††] | 100 | 2 | 7.00 ± 0.27 | 0.9991 | 90 ± 13 | 0.9992 | 4 | 102 ± 5 | 0.9999 |
| WT-pF1 (F569[S3]A) | 379 | 6 | 6.42 ± 0.13 | 0.1397 | 108 ± 7 | 0.9995 | 3 | 100 | / |
| WT-pF1 (L572[S6]A) | 678 | 11 | 6.17 ± 0.22* | 0.0064 | 101 ± 12 | >0.9999 | 3 | 100 | / |
| WT-pF1 (M573[S7]A) | 504 | 8 | 6.30 ± 0.23* | 0.0354 | 87 ± 11 | 0.9991 | 3 | 100 | / |

Stalk binding pocket

| Mutants[†] | $EC_{50}$ (nM) | $EC_{50}$ ratio[‡] | pEC50 mean ± s.e.m.[§] | P value | $E_{max}$[‖] % of WT[§] | P value | n[¶] | Expression[#] % of WT | P value |
|---|---|---|---|---|---|---|---|---|---|
| V585[1.39b]A | 436 | 7 | 6.36 ± 0.19 | 0.0724 | 84 ± 8 | 0.9987 | 3 | 91 ± 21 | 0.9993 |
| T589[1.43b]A | 87 | 1 | 7.06 ± 0.21 | 0.9995 | 114 ± 13 | 0.9990 | 3 | 83 ± 3 | 0.9922 |
| L593[1.47b]A | 559 | 9 | 6.25 ± 0.21* | 0.0188 | 106 ± 11 | 0.9996 | 3 | 90 ± 6 | 0.9991 |
| F641[2.64b]A | 202 | 3 | 6.69 ± 0.32 | 0.8802 | 82 ± 15 | 0.9984 | 3 | 82 ± 19 | 0.9828 |
| Y668[3.40b]A | 356 | 6 | 6.45 ± 0.26 | 0.1890 | 123 ± 17 | 0.9709 | 3 | 67 ± 1 | 0.1270 |
| V732[ECL2]A | 1,946 | 32 | 5.71 ± 0.26*** | <0.0001 | 73 ± 13 | 0.8356 | 3 | 91 ± 21 | 0.9993 |
| W734[ECL2]A | 138 | 2 | 6.86 ± 0.37 | 0.9985 | 60 ± 11 | 0.1282 | 3 | 87 ± 1 | 0.9987 |
| F747[5.39b]A | 1,218 | 20 | 5.91 ± 0.27** | 0.0001 | 80 ± 13 | 0.9925 | 3 | 96 ± 1 | 0.9997 |
| V748[5.40b]A | 57 | 1 | 7.25 ± 0.34 | 0.9999 | 85 ± 17 | 0.9988 | 3 | 99 ± 9 | >0.9999 |
| L752[5.44b]A | 511 | 8 | 6.29 ± 0.31* | 0.0313 | 90 ± 15 | 0.9993 | 3 | 105 ± 6 | 0.9997 |
| W804[6.53b]A | 984 | 16 | 6.01 ± 0.36*** | <0.0001 | 86 ± 19 | 0.9987 | 3 | 93 ± 4 | 0.9995 |
| G807[6.56b]W | 1,525 | 25 | 5.82 ± 0.26*** | <0.0001 | 95 ± 15 | 0.9997 | 3 | 101 ± 21 | >0.9999 |
| T810[6.59b]A | 566 | 9 | 6.25 ± 0.25* | 0.0188 | 81 ± 11 | 0.9982 | 3 | 92 ± 8 | 0.9994 |
| H820[7.39b]A | 274 | 5 | 6.56 ± 0.19 | 0.4778 | 126 ± 12 | 0.8827 | 3 | 77 ± 1 | 0.7768 |
| F823[7.42b]A | 317 | 5 | 6.50 ± 0.30 | 0.2998 | 90 ± 14 | 0.9993 | 3 | 22 ± 2*** | <0.0001 |
| N827[7.46b]A | 391 | 6 | 6.41 ± 0.19 | 0.1258 | 122 ± 13 | 0.9824 | 3 | 97 ± 21 | 0.9998 |

Signaling cascade

| Mutants[†] | $EC_{50}$ (nM) | $EC_{50}$ ratio[‡] | pEC50 mean ± s.e.m.[§] | P value | $E_{max}$[‖] % of WT[§] | P value | n[¶] | Expression[#] % of WT | P value |
|---|---|---|---|---|---|---|---|---|---|
| F672[3.44b]A | 638 | 10 | 6.20 ± 0.31* | 0.0097 | 72 ± 12 | 0.7805 | 3 | 101 ± 2 | >0.9999 |
| M675[3.47b]A | 425 | 7 | 6.37 ± 0.31 | 0.0811 | 81 ± 13 | 0.9982 | 3 | 107 ± 2 | 0.9995 |
| V755[5.47b]A | nd | nd | nd | nd | nd | nd | 3 | 111 ± 15 | 0.9990 |
| W804[6.53b]A | 984 | 16 | 6.01 ± 0.36*** | <0.0001 | 86 ± 19 | 0.9987 | 3 | 93 ± 4 | 0.9995 |
| Q830[7.49b]A | 847 | 14 | 6.07 ± 0.22* | 0.0015 | 124 ± 15 | 0.9499 | 3 | 115 ± 8 | 0.9985 |
| G831[7.50b]A | 1,035 | 17 | 5.99 ± 0.26** | 0.0005 | 97 ± 15 | 0.9998 | 3 | 70 ± 16 | 0.2935 |
| M627[2.50b]A | 696 | 11 | 6.16 ± 0.12* | 0.0056 | 113 ± 8 | 0.9991 | 3 | 101 ± 11 | >0.9999 |
| L678[3.50b]A | 129 | 2 | 6.89 ± 0.23 | 0.9987 | 58 ± 7 | 0.0828 | 3 | 52 ± 1* | 0.0013 |
| P798[6.47b]A | 920 | 15 | 6.04 ± 0.38* | 0.0010 | 74 ± 15 | 0.8827 | 3 | 104 ± 33 | 0.9997 |
| L799[6.48b]A | 382 | 6 | 6.42 ± 0.19 | 0.1397 | 109 ± 11 | 0.9994 | 3 | 120 ± 21 | 0.9606 |
| L800[6.49b]A | 255 | 4 | 6.59 ± 0.15 | 0.5796 | 105 ± 8 | 0.9997 | 3 | 105 ± 4 | 0.9997 |
| G801[6.50b]A | 346 | 6 | 6.46 ± 0.19 | 0.2083 | 127 ± 13 | 0.8356 | 3 | 97 ± 12 | 0.9998 |
| L681[3.53b]A | 119 | 2 | 6.92 ± 0.32 | 0.9989 | 59 ± 10 | 0.1034 | 3 | 93 ± 3 | 0.9994 |
| L681[3.53b]H | 345 | 6 | 6.46 ± 0.22 | 0.0698 | 103 ± 13 | 0.9998 | 4 | 98 ± 5 | 0.9999 |
| L681[3.53b]D | 791 | 13 | 6.10 ± 0.17* | 0.0024 | 130 ± 12 | 0.6523 | 3 | 125 ± 19 | 0.6699 |
| L682[3.54b]A | 268 | 4 | 6.57 ± 0.27 | 0.5112 | 79 ± 12 | 0.9853 | 3 | 93 ± 2 | 0.9995 |
| A683[3.55b]G | 335 | 6 | 6.47 ± 0.22 | 0.2291 | 111 ± 13 | 0.9993 | 3 | 116 ± 9 | 0.9933 |
| V761[5.53b]A | 1,624 | 27 | 5.79 ± 0.27*** | <0.0001 | 124 ± 22 | 0.9499 | 3 | 105 ± 19 | 0.9997 |
| L764[5.56b]A | 1,299 | 21 | 5.89 ± 0.28*** | <0.0001 | 95 ± 16 | 0.9997 | 3 | 111 ± 22 | 0.9991 |
| V765[5.57b]A | 1,001 | 16 | 6.00 ± 0.29** | 0.0005 | 92 ± 16 | 0.9995 | 3 | 113 ± 26 | 0.9987 |
| L796[6.45b]A | 562 | 9 | 6.25 ± 0.48* | 0.0188 | 35 ± 9** | 0.5434 | 3 | 74 ± 10 | 0.5434 |

Lipid pocket

| Mutants[†] | $EC_{50}$ (nM) | $EC_{50}$ ratio[‡] | pEC50 mean ± s.e.m.[§] | P value | $E_{max}$[‖] % of WT[§] | P value | n[¶] | Expression[#] % of WT | P value |
|---|---|---|---|---|---|---|---|---|---|
| Y684[3.56b]A | 814 | 13 | 6.09 ± 0.26* | 0.0021 | 115 ± 17 | 0.9988 | 3 | 103 ± 8 | 0.9998 |
| G702[4.45b]Y | 1,175 | 19 | 5.93 ± 0.29** | 0.0002 | 94 ± 16 | 0.9996 | 3 | 87 ± 12 | 0.9988 |

G protein binding pocket

| Mutants[†] | $EC_{50}$ (nM) | $EC_{50}$ ratio[‡] | pEC50 mean ± s.e.m.[§] | P value | $E_{max}$[‖] % of WT[§] | P value | n[¶] | Expression[#] % of WT | P value |
|---|---|---|---|---|---|---|---|---|---|
| R623[2.46b]A | 444 | 7 | 6.35 ± 0.13 | 0.0645 | 120 ± 8 | 0.9925 | 3 | 118 ± 5 | 0.9846 |
| V689[ICL2]A | 263 | 4 | 6.58 ± 0.21 | 0.5452 | 67 ± 7 | 0.4502 | 3 | 112 ± 3 | 0.9989 |
| F690[ICL2]A | 80 | 1 | 7.10 ± 0.29 | 0.9997 | 94 ± 15 | 0.9996 | 3 | 102 ± 9 | 0.9999 |
| L769[5.61b]A | 301 | 5 | 6.52 ± 0.22 | 0.3540 | 102 ± 12 | 0.9999 | 3 | 100 ± 6 | >0.9999 |
| R788[6.37b]A | nd | nd | nd | nd | nd | nd | 3 | 92 ± 4 | 0.9994 |
| K791[6.40b]A | 102 | 2 | 6.99 ± 0.23 | 0.9993 | 102 ± 13 | 0.9999 | 3 | 103 ± 12 | 0.9998 |

### pD1-induced $G_s$ activation of ADGRD1

| Mutants[†] | $EC_{50}$ (nM) | $EC_{50}$ ratio[‡] | pEC50 mean ± s.e.m.[§] | P value | $E_{max}$[‖] % of WT[§] | P value | n[¶] | Expression[#] % of WT | P value |
|---|---|---|---|---|---|---|---|---|---|
| WT | 22 | 1 | 7.66 ± 0.08 | / | 100 ± 3 | / | 22 | 100 | / |
| WT-pF1 | 1,270 | 59 | 5.90 ± 0.29*** | <0.0001 | 67 ± 12 | 0.2088 | 3 | 100 | / |
| Construct[††] | 52 | 2 | 7.28 ± 0.29 | 0.9942 | 98 ± 13 | 0.9999 | 3 | 34 ± 3*** | <0.0001 |
| WT-pD1 (F547[S3]A) | 721 | 33 | 6.14 ± 0.38*** | <0.0001 | 69 ± 13 | 0.3077 | 3 | 100 | / |
| WT-pD1 (L550[S6]A) | 166 | 8 | 6.78 ± 0.21 | 0.1190 | 53 ± 5* | 0.0055 | 3 | 100 | / |
| WT-pD1 (M551[S7]A) | 262 | 12 | 6.58 ± 0.30* | 0.0146 | 79 ± 10 | 0.9320 | 3 | 100 | / |
| Q563[1.36b]A | 1,096 | 51 | 5.96 ± 0.25*** | <0.0001 | 106 ± 13 | 0.9996 | 3 | 133 ± 22 | 0.1307 |
| L566[1.39b]A | 1,219 | 56 | 5.91 ± 0.20*** | <0.0001 | 107 ± 10 | 0.9995 | 3 | 96 ± 19 | 0.9997 |
| S570[1.43b]A | 93 | 4 | 7.03 ± 0.35 | 0.4429 | 80 ± 12 | 0.8635 | 4 | 121 ± 2 | 0.9086 |
| L619[2.64b]A | 980 | 45 | 6.01 ± 0.36*** | <0.0001 | 75 ± 13 | 0.7202 | 3 | 91 ± 10 | 0.9992 |
| F643[3.40b]A | 647 | 30 | 6.19 ± 0.21*** | <0.0001 | 89 ± 8 | 0.9991 | 3 | 101 ± 10 | >0.9999 |
| N703[ECL2]A | 960 | 44 | 6.02 ± 0.15*** | <0.0001 | 118 ± 9 | 0.9850 | 3 | 103 ± 4 | 0.9997 |
| W705[ECL2]A | 23 | 1 | 7.64 ± 0.25 | >0.9999 | 77 ± 7 | 0.8449 | 3 | 64 ± 7 | 0.0617 |
| F716[5.39b]A | 258 | 12 | 6.59 ± 0.32* | 0.0164 | 80 ± 11 | 0.9599 | 3 | 98 ± 3 | 0.9999 |
| V717[5.40b]A | 80 | 4 | 7.10 ± 0.19 | 0.6750 | 98 ± 8 | 0.9999 | 4 | 109 ± 7 | 0.9992 |
| W773[6.53b]A | nd | nd | nd | nd | nd | nd | 3 | 69 ± 2 | 0.2383 |
| Q788[7.39b]A | 38 | 2 | 7.42 ± 0.26 | 0.9992 | 110 ± 11 | 0.9992 | 3 | 75 ± 11 | 0.5967 |
| F791[7.42b]A | nd | nd | nd | nd | nd | nd | 3 | 53 ± 6* | 0.0025 |
| N795[7.46b]A | 14 | 0.6 | 7.86 ± 0.20 | 0.9994 | 95 ± 9 | 0.9996 | 3 | 73 ± 6 | 0.4814 |
| F647[3.44b]A | 1,014 | 47 | 5.99 ± 0.32*** | <0.0001 | 64 ± 10 | 0.1080 | 3 | 60 ± 10* | 0.0232 |
| M650[3.47b]A | 200 | 9 | 6.70 ± 0.25 | 0.0543 | 99 ± 10 | >0.99993 | 4 | 68 ± 4 | 0.1751 |
| I724[5.47b]A | 7,052 | 325 | 5.15 ± 0.22*** | <0.0001 | 113 ± 14 | 0.9988 | 3 | 113 ± 20 | 0.9986 |
| W773[6.53b]A | nd | nd | nd | nd | nd | nd | 3 | 69 ± 2 | 0.2383 |
| Q798[7.49b]A | 180 | 8 | 6.75 ± 0.32 | 0.0894 | 96 ± 13 | 0.9997 | 3 | 96 ± 18 | 0.9997 |
| G799[7.50b]A | 339 | 16 | 6.47 ± 0.37* | 0.0040 | 66 ± 10 | 0.1693 | 3 | 69 ± 3 | 0.2134 |
| H605[2.50b]A | 2,433 | 112 | 5.61 ± 0.27*** | <0.0001 | 113 ± 14 | 0.9988 | 3 | 102 ± 7 | 0.9999 |
| E653[3.50b]A | 4,538 | 209 | 5.34 ± 0.33*** | <0.0001 | 102 ± 17 | 0.9999 | 4 | 92 ± 25 | 0.9993 |
| P767[6.47b]A | 759 | 35 | 6.12 ± 0.27*** | <0.0001 | 75 ± 9 | 0.7202 | 3 | 90 ± 6 | 0.9991 |
| I768[6.48b]A | 47 | 2 | 7.33 ± 0.31 | 0.9988 | 69 ± 9 | 0.3077 | 3 | 60 ± 2* | 0.0242 |
| L769[6.49b]A | nd | nd | nd | nd | nd | nd | 4 | 90 ± 12 | 0.9991 |
| G770[6.50b]A | 2,347 | 108 | 5.63 ± 0.33*** | <0.0001 | 83 ± 14 | 0.9927 | 3 | 131 ± 1 | 0.2483 |
| H656[3.53b]A | 1,708 | 79 | 5.77 ± 0.33*** | <0.0001 | 87 ± 14 | 0.9988 | 3 | 108 ± 2 | 0.9993 |
| H656[3.53b]D | 686 | 32 | 6.16 ± 0.30*** | <0.0001 | 90 ± 16 | 0.9990 | 4 | 77 ± 28 | 0.7768 |
| L657[3.54b]A | 2,233 | 103 | 5.65 ± 0.24*** | <0.0001 | 69 ± 8 | 0.3077 | 3 | 28 ± 1*** | <0.0001 |
| Y658[3.55b]A | 408 | 19 | 6.39 ± 0.25* | 0.0014 | 85 ± 10 | 0.9985 | 3 | 110 ± 9 | 0.9992 |
| I730[5.53b]A | 904 | 42 | 6.04 ± 0.28*** | <0.0001 | 67 ± 9 | 0.2088 | 3 | 76 ± 5 | 0.7231 |
| V734[5.57b]A | 359 | 17 | 6.45 ± 0.26* | 0.0031 | 122 ± 13 | 0.8938 | 3 | 134 ± 24 | 0.1259 |
| L765[6.45b]A | 786 | 36 | 6.10 ± 0.22*** | <0.0001 | 75 ± 8 | 0.7202 | 3 | 84 ± 19 | 0.9929 |
| R598[ICL1]A | 350 | 16 | 6.46 ± 0.26* | 0.0035 | 97 ± 12 | 0.9998 | 3 | 97 ± 6 | 0.9998 |
| R601[2.46b]A | 336 | 16 | 6.47 ± 0.19* | 0.0040 | 107 ± 8 | 0.9995 | 3 | 104 ± 5 | 0.9997 |
| V664[ICL2]A | 195 | 9 | 6.71 ± 0.33 | 0.0601 | 71 ± 9 | 0.4321 | 3 | 106 ± 6 | 0.9995 |
| F665[ICL2]A | 876 | 40 | 6.06 ± 0.25*** | <0.0001 | 66 ± 8 | 0.1693 | 3 | 116 ± 5 | 0.9925 |
| K760[6.40b]A | 10,186 | 470 | 4.99 ± 0.33*** | <0.0001 | 77 ± 15 | 0.8449 | 3 | 126 ± 21 | 0.5264 |
| V764[6.44b]A | 3,244 | 150 | 5.49 ± 0.30*** | <0.0001 | 85 ± 13 | 0.9985 | 3 | 82 ± 4 | 0.9735 |

### pF1-induced $G_s$ activation of ADGRF1

| Mutants[†] | $EC_{50}$ (nM) | $EC_{50}$ ratio[‡] | pEC50 mean ± s.e.m.[§] | P value | $E_{max}$[‖] % of WT[§] | P value | n[¶] | Expression[#] % of WT | P value |
|---|---|---|---|---|---|---|---|---|---|
| WT | 36 | 1 | 7.45 ± 0.15 | / | 100 ± 9 | / | 7 | 100 | / |
| R623[2.46b]A | 147 | 4 | 6.83 ± 0.29 | 0.2432 | 57 ± 9 | 0.0517 | 3 | 116 ± 10 | 0.5574 |
| V689[ICL2]A | 205 | 6 | 6.69 ± 0.25 | 0.1046 | 77 ± 10 | 0.5261 | 3 | 104 ± 11 | 0.9981 |
| F690[ICL2]A | 114 | 3 | 6.94 ± 0.20 | 0.4285 | 110 ± 12 | 0.9755 | 3 | 92 ± 1 | 0.9574 |
| L769[5.61b]A | 152 | 4 | 6.82 ± 0.30 | 0.2299 | 73 ± 11 | 0.3638 | 3 | 111 ± 13 | 0.8497 |
| R788[6.37b]A | 938 | 26 | 6.03 ± 0.23* | 0.0010 | 112 ± 15 | 0.9441 | 3 | 104 ± 6 | 0.9985 |
| K791[6.40b]A | 909 | 25 | 6.04 ± 0.29* | 0.0011 | 79 ± 13 | 0.6155 | 3 | 113 ± 10 | 0.7312 |

(G protein binding pocket)

### A8-induced $G_s$ activation of ADGRF1

| Mutants[†] | $EC_{50}$ (nM) | $EC_{50}$ ratio[‡] | pEC50 mean ± s.e.m.[§] | P value | $E_{max}$[‖] % of WT[§] | P value | n[¶] | Expression[#] % of WT | P value |
|---|---|---|---|---|---|---|---|---|---|
| WT | 1.1 | 1 | 8.97 ± 0.11 | / | 100 ± 5 | / | 13 | 100 | / |
| CTF | 3.9 | 4 | 8.41 ± 0.20 | 0.1646 | 116 ± 10 | 0.5145 | 4 | 46 ± 2*** | <0.0001 |
| TMD | 3.8 | 4 | 8.42 ± 0.24 | 0.1767 | 113 ± 11 | 0.6884 | 4 | 53 ± 8*** | <0.0001 |
| Y684[3.56b]A | 32 | 29 | 7.50 ± 0.27*** | <0.0001 | 89 ± 10 | 0.7030 | 6 | 106 ± 6 | 0.4465 |
| G702[4.45b]Y | 65 | 59 | 7.19 ± 0.21*** | <0.0001 | 99 ± 10 | 0.9999 | 4 | 100 ± 2 | 0.9998 |

(Lipid binding)

### A8-induced $G_i$ activation of ADGRF1

| Mutants[†] | $EC_{50}$ (nM) | $EC_{50}$ ratio[‡] | pEC50 mean ± s.e.m.[§] | P value | $E_{max}$[‖] % of WT[§] | P value | n[¶] | Expression[#] % of WT | P value |
|---|---|---|---|---|---|---|---|---|---|
| WT | 2.0 | 1 | 8.70 ± 0.07 | / | 100 ± 3 | / | 13 | 100 | / |
| CTF | 2.1 | 1 | 8.68 ± 0.11 | 0.9998 | 93 ± 7 | 0.7170 | 4 | 66 ± 4*** | <0.0001 |
| TMD | 2.6 | 1 | 8.59 ± 0.13 | 0.9179 | 91 ± 5 | 0.5109 | 4 | 70 ± 7** | 0.0001 |
| Y684[3.56b]A | 95 | 48 | 7.02 ± 0.16*** | <0.0001 | 95 ± 7 | 0.8587 | 5 | 101 ± 12 | 0.9997 |
| G702[4.45b]Y | 63 | 32 | 7.20 ± 0.13*** | <0.0001 | 90 ± 5 | 0.3383 | 5 | 94 ± 1 | 0.7237 |

[†]All mutations were introduced in the wild-type receptor.

[‡]The $EC_{50}$ ratio ($EC_{50(mutant)}/EC_{50(WT)}$) represents the shift between the wild-type and mutant curves, and characterizes the effect of the mutations on G protein activation.

[§]Data are mean ± s.e.m. from at least three independent experiments. *$P < 0.05$, **$P < 0.001$, ***$P < 0.0001$ by one-way analysis of variance followed by Dunnett's post-test compared to the response of wild type. nd, not determined (data for which the concentration response curve could not reach effect saturation within the concentration range tested).

[‖]The maximal response is reported as a percentage of the maximum effect at the wild type.

[¶]Sample size, the number of independent experiments performed in technical duplicate.

[#]Protein expression levels of ADGRD1 and ADGRF1 constructs at the cell surface were determined in parallel by flow cytometry with an anti-Flag antibody (Sigma) and reported as per cent compared to the wild type from three independent measurements performed in duplicate. The mutants with low expression level (less than 40% of wild-type expression level) are indicated with a grey background. [††]The ADGRD1 and ADGRF1 constructs that were used to determine the structures. See Extended Data Fig. 1a for schematic diagrams of the constructs.

# Reporting Summary

## Statistics

For all statistical analyses, confirm that the following items are present in the figure legend, table legend, main text, or Methods section.

| n/a | Confirmed | |
|---|---|---|
| ☐ | ☒ | The exact sample size (*n*) for each experimental group/condition, given as a discrete number and unit of measurement |
| ☐ | ☒ | A statement on whether measurements were taken from distinct samples or whether the same sample was measured repeatedly |
| ☐ | ☒ | The statistical test(s) used AND whether they are one- or two-sided *Only common tests should be described solely by name; describe more complex techniques in the Methods section.* |
| ☒ | ☐ | A description of all covariates tested |
| ☒ | ☐ | A description of any assumptions or corrections, such as tests of normality and adjustment for multiple comparisons |
| ☐ | ☒ | A full description of the statistical parameters including central tendency (e.g. means) or other basic estimates (e.g. regression coefficient) AND variation (e.g. standard deviation) or associated estimates of uncertainty (e.g. confidence intervals) |
| ☐ | ☒ | For null hypothesis testing, the test statistic (e.g. *F*, *t*, *r*) with confidence intervals, effect sizes, degrees of freedom and *P* value noted *Give P values as exact values whenever suitable.* |
| ☒ | ☐ | For Bayesian analysis, information on the choice of priors and Markov chain Monte Carlo settings |
| ☒ | ☐ | For hierarchical and complex designs, identification of the appropriate level for tests and full reporting of outcomes |
| ☒ | ☐ | Estimates of effect sizes (e.g. Cohen's *d*, Pearson's *r*), indicating how they were calculated |

*Our web collection on statistics for biologists contains articles on many of the points above.*

## Software and code

Policy information about availability of computer code

| Data collection | Automated data collection on the Titan Krios was performed using serialEM 3.7. |
|---|---|
| Data analysis | The following softwares were used in cryo-EM data processing, model building, and structure validation: MotionCor2 v1.4.2, Gctf v1.18, RELION3.1, ResMap v1.1.4, ChimeraX v.1.1, COOT 0.8.9, PHENIX 1.19.2, and MolProbity 4.2. The functional data were analyzed by GraphPad Prism 8.0. The mass spectrometry data were analyzed using MS-DIAL 4.70 and TraceFinder 4.0. The figures were prepared using PyMOL 1.8 and UCSF Chimera 1.15. |

For manuscripts utilizing custom algorithms or software that are central to the research but not yet described in published literature, software must be made available to editors and reviewers. We strongly encourage code deposition in a community repository (e.g. GitHub). See the Nature Portfolio guidelines for submitting code & software for further information.

## Data

Policy information about availability of data

All manuscripts must include a data availability statement. This statement should provide the following information, where applicable:

- Accession codes, unique identifiers, or web links for publicly available datasets
- A description of any restrictions on data availability
- For clinical datasets or third party data, please ensure that the statement adheres to our policy

Atomic coordinates and cryo-EM density maps for the structures of ADGRD1–miniGs, ADGRF1–miniGs, ADGRF1–miniGi1 and ADGRF1(H565A/T567A)–miniGi1 complexes have been deposited in the PDB under identification codes 7WU2, 7WU3, 7WU4 and 7WU5, respectively, and in the Electron Microscopy Data Bank under accession codes EMD-32817, EMD-32818, EMD-32819 and EMD-32820, respectively.

# Field-specific reporting

Please select the one below that is the best fit for your research. If you are not sure, read the appropriate sections before making your selection.

☒ Life sciences ☐ Behavioural & social sciences ☐ Ecological, evolutionary & environmental sciences

For a reference copy of the document with all sections, see nature.com/documents/nr-reporting-summary-flat.pdf

# Life sciences study design

All studies must disclose on these points even when the disclosure is negative.

| | |
|---|---|
| Sample size | No statistical methods were used to predetermine sample size. All functional data were obtained from at least three independent experiments to ensure each data point was repeatable and comparable to other published studies. Wild-type receptors were tested in parallel as controls with a large number of repeats. Sample size for the cryo-EM studies was determined by availability of microscope time and to ensure unambiguous modeling of most of residues that allowed us to obtain a high-resolution reconstruction. |
| Data exclusions | No data were excluded from the analyses. |
| Replication | All functional assays were performed in technical triplicate or duplicate. All attempts at replication were successful. |
| Randomization | Randomization is not relevant to this study, as all experiments did not allocate experimental groups. |
| Blinding | Blinding is not relevant to this study, as no subjective allocation was involved in any of the structural and functional experiments. |

# Reporting for specific materials, systems and methods

We require information from authors about some types of materials, experimental systems and methods used in many studies. Here, indicate whether each material, system or method listed is relevant to your study. If you are not sure if a list item applies to your research, read the appropriate section before selecting a response.

### Materials & experimental systems

| n/a | Involved in the study |
|---|---|
| ☐ | ☒ Antibodies |
| ☐ | ☒ Eukaryotic cell lines |
| ☒ | ☐ Palaeontology and archaeology |
| ☒ | ☐ Animals and other organisms |
| ☒ | ☐ Human research participants |
| ☒ | ☐ Clinical data |
| ☒ | ☐ Dual use research of concern |

### Methods

| n/a | Involved in the study |
|---|---|
| ☒ | ☐ ChIP-seq |
| ☐ | ☒ Flow cytometry |
| ☒ | ☐ MRI-based neuroimaging |

## Antibodies

| | |
|---|---|
| Antibodies used | Cryptate-labelled anti-IP1 monoclonal antibody: CisBio Bioassays, Cat#62IPAPEC, 1:20 diluted in lysis and detection buffer; IP1-d2 antibody: Cisbio Bioassays, Cat#62IPAPEC, 1:20 in lysis and detection buffer; anti-FLAG antibody: Sigma, Cat#F4049, 1:120 diluted in TBS supplemented with 4% BSA and 20% viability staining solution 7-AAD (Invitrogen). |
| Validation | All antibodies were commercially obtained and validation reports are available on the supplier website: Cryptate-labelled anti-IP1 monoclonal antibody: https://www.cisbio.cn/ip-one-gq-kit-40451#section-products-tabs-product; IP1-d2 antibody: https://www.cisbio.cn/media/asset/l/s/ls-pr-a-new-inositol-phosphate-assay-to-monitor-gq-coupled-gpcr-responses-a-functional-assay-to-monitor-the-activation-of-gq-coupled-receptors-in-a-hts-format.pdf; anti-FLAG antibody: https://www.sigmaaldrich.com/technical-documents/articles/biofiles/antibodies-to-peptides.html. |

## Eukaryotic cell lines

Policy information about cell lines

| | |
|---|---|
| Cell line source(s) | The High Five and HEK293F cell lines were originally obtained from Invitrogen. |
| Authentication | None of the cell lines have been authenticated. |
| Mycoplasma contamination | The cell lines were negative for mycoplasma contamination. |

| Commonly misidentified lines<br>(See ICLAC register) | No commonly misidentified cell lines were used. |
|---|---|

# Flow Cytometry

## Plots

Confirm that:

☐ The axis labels state the marker and fluorochrome used (e.g. CD4-FITC).

☐ The axis scales are clearly visible. Include numbers along axes only for bottom left plot of group (a 'group' is an analysis of identical markers).

☐ All plots are contour plots with outliers or pseudocolor plots.

☐ A numerical value for number of cells or percentage (with statistics) is provided.

## Methodology

| Sample preparation | Cell surface expression of the receptors was measured by incubating 10 ul cells with 15 ul monoclonal anti-Flag M2-FITC antibody (Sigma; 1:120 diluted in TBS supplemented with 4% BSA and 20% viability staining solution 7-AAD (Invitrogen)) at 4 °C for 20 min. After incubation, 175 ul TBS buffer was added and the fluorescent signal was measured using a flow cytometry reader (Guava easyCyte HT, Millipore). |
|---|---|
| Instrument | Guava easyCyte HT, Millipore |
| Software | The data were collected and analyzed by GuavaSoft 2.2.2, Guava ExpressPlus panel. |
| Cell population abundance | For each measurement, 2,000 cell events were collected and the fluorescence intensity of cell population with protein expression was calculated. |
| Gating strategy | Gating was determined by the Green-red fluorescence intensity to differentiate positive cells. |

☐ Tick this box to confirm that a figure exemplifying the gating strategy is provided in the Supplementary Information.

