## [Peer Review File · Nature]

Manuscript Title: Structural basis of tethered agonism of the adhesion GPCRs ADGRD1 and ADGRF1

Reviewer Comments & Author Rebuttals

Reviewer Reports on the Initial Version:

Referees' comments:

Referee #1 (Remarks to the Author):

Reviewer's comments

In a well-written report, the authors of "Structural basis of tethered agonism of the adhesion GPCRs ADGRD1 and ADGRF1" describe the structural determination by cryo-EM of two aGPCRs members from 2 different families coupled to their cognate G proteins. While the field of GPCR has seen a plethora of resolved structures published in the last 10 years aided by important methodological improvements, the branch corresponding to adhesion GPCRs has remained consistently unexplored. A recent breakthrough study (Ping et al., Nature, 2021) did not contemplate the description of a cryptic ligand known to be modulating receptor activity in the majority of aGPCRs. Decryption of such a ligand was thought to result from an autoproteolysis event followed by the dissociation of the then generated subunits allowing for the exposition of a tethered sequence. While mechanistic models supporting the activation of aGPCRs are still controversial, all models have uniformly settled on the importance of this cryptic ligand to convey extracellular stimuli into intracellular signaling cascades. A lingering question at the center of this conundrum has been how the cryptic ligand act in order to imprint a conformational change to the seven-transmembrane domain leading to intracellular signaling. This report brings crucial evidence supporting the engulfment of the cryptic ligand into a binding pocket delimited by the transmembrane domain and extracellular loops, thus bringing a clear structural correlate to previous biochemical studies. Moreover, the authors go on to show that the decryption of the tethered ligand can occur independently from the auto-proteolysis event and that the ligand adopts a similar binding conformation as the proteolytically-decrypting ligand, which might come as a surprise to the aGPCR field. Finally, the authors pioneer the identification of a lipid binding site for ADGRF1 not present in ADGRD1 and which seems to regulate receptor activity. Through a series of mutational analysis, the authors provide further support to their cryo-EM determination by relating receptor activity to intramolecular interactions surrounding the embedded tethered ligand and lipid binding site. Finally, by describing two different G protein coupling modes for the same aGPCR, the authors widen our understanding of structural determinants driving receptor-mediated G protein signaling selectivity. In summary, this report allows for a unique cross-comparison of receptor/tethered ligand/G protein interface in an attempt to unify aGPCR structure-function relationship.

While this reviewer finds this report to be nearly complete, a few minor details need to be addressed in a revised version of the manuscript:

1. Methods section: Please indicate the selected human variants of sequences used as templates for ADGRD1 and ADGRF1 mutants and WT
2. Methods section: Please indicate which molecular biology method was used to generate receptor mutations
3. Methods section: For BRET assays with stalk peptide, please indicate how the synthetic ligands were solubilized in order to be used in these assays (solvent, stock concentration, etc)
4. Methods section: Specify the exact biosensors per G protein family used in the methods and the corresponding conditions for transfection assays (which set of G protein subunits for each family of Gi and Gs were used, DNA amount used for each component including receptor plasmids, number of cells transfected per sample with biosensors and receptor DNA)
5. Methods section: Please indicate which DNA amount were used in order to obtain the constitutive activity described in cAMP and IP accumulation assays
6. Methods section: It is unclear how the synthetic stalk peptides were obtained or generated. Please describe.
7. Please provide a schematic of pF1 and pD1 -derived peptides used in the study
8. Fig 2 i,j: the figures indicates LS3 but should read LS6
9. Fig 2 i,j: For ease of reading, please reiterate/indicate the color legend attributed to either the stalk or receptor (non-stalk region)
10. Fig 2 i,j: please indicate stalk residues facing outward or inward beside the respective "Stalk-N" headers (Maybe "Stalk-N inward" and "Stalk-N outward")
11. Fig 2g: Please include missing residue T810 in the ribbon structure
12. Fig 2d, figure legend: please provide a short explanation of the mechanism for stalk-TMD interaction in absence of GAIN domain dissociation
13. Statistics: Given that the data in Fig 2i,j/Fig 3h,i/Fig 4c/were calculated from mean values, the variation should also be present in control samples used as a normalization reference. Please provide error bars corresponding to control samples.
14. Extended data Fig 5 and extended data table 3: It is unclear how the authors were able to determine EC50 values for curves which did not reach effect saturation. Please clarify in rebuttal and include the relevant information as part of extended data table 3 legend.
15. Quantification of protein expression seems inaccurate in a few cases. For example, F641A mutant is reported at 19% of WT expression in extended data table 2 but at 82 % in extended data

table 3, using the same flow cytometry protocol... Please revise these inconsistencies accordingly in the manuscript.

Referee #2 (Remarks to the Author):

The manuscript by Qu and colleagues reports cryo-EM structures of two adhesion GPCRs, the ADGRD1 and the ADGRF1. While significant progress has been made over the past few years on structural understanding of GPCR function, the molecular basis of action of adhesion GPCRs remain poorly understood. The authors determined 4 cryo-EM structures of these two receptors to decipher how adhesion receptors are activated. A central question for the field has been how a tethered agonist, released by autoproteolysis of the GAIN domain, activates the 7TM domain of an adhesion receptor. Especially confusing has been whether proteolysis is strictly required for adhesion GPCR activation, as several lines of evidence suggest that this is not needed per se. The major success of the authors work is that it reveals the active conformation of two adhesion receptors, in one case the same receptor bound to two engineered miniG proteins, thereby providing some insight into differential G protein coupling. Perhaps the most interesting result is that a construct that cannot undergo proteolysis shows a tethered agonist conformation similar to the situation without the N-terminal fragment. The major weakness of the present manuscript is that it is written in a style where the figures and text largely report a laundry list of interactions, which is likely to be less interesting to a generalist audience. Regardless, the work is well done and thorough, with solid structural biology and functional studies.

While much of the mutagenesis proposed by the authors is useful, it largely confirms the interactions observed in the structures and provides little extra perspective for the field. I instead suggest that the authors revise the manuscript and put into context their very interesting observation that the uncleaved ADGRF1 is still in a similar conformation as the construct without the NTF. There is much to unpack here - is the expectation that activation leads to unfolding of the GAIN domain? Is there a stimulus required for this to happen? Do these constructs have a difference in constitutive activity; if so, how does one reconcile that with the structural observation?

Author Rebuttals to Initial Comments:

Response to referee comments

Referee #1 (Remarks to the Author):

In a well-written report, the authors of “Structural basis of tethered agonism of the adhesion GPCRs ADGRD1 and ADGRF1” describe the structural determination by cryo-EM of two aGPCRs members from 2 different families coupled to their cognate G proteins. While the field of GPCR has seen a plethora of resolved structures published in the last 10 years aided by important methodological improvements, the branch corresponding to adhesion GPCRs has remained consistently unexplored. A recent breakthrough study (Ping et al., Nature, 2021) did not contemplate the description of a cryptic ligand known to be modulating receptor activity in the majority of aGPCRs. Decryption of such a ligand was thought to result from an autoproteolysis event followed by the dissociation of the then generated subunits allowing for the exposition of a tethered sequence. While mechanistic models supporting the activation of aGPCRs are still controversial, all models have uniformly settled on the importance of this cryptic ligand to convey extracellular stimuli into intracellular signaling cascades. A lingering question at the center of this conundrum has been how the cryptic ligand act in order to imprint a conformational change to the seven-transmembrane domain leading to intracellular signaling. This report brings crucial evidence supporting the engulfment of the cryptic ligand into a binding pocket delimited by the transmembrane domain and extracellular loops, thus bringing a clear structural correlate to previous biochemical studies. Moreover, the authors go on to show that the decryption of the tethered ligand can occur independently from the auto-proteolysis event and that the ligand adopts a similar binding conformation as the proteolytically-decrypting ligand, which might come as a surprise to the aGPCR field. Finally, the authors pioneer the identification of a lipid binding site for ADGRF1 not present in ADGRD1 and which seems to regulate receptor activity. Through a series of mutational analysis, the authors provide further support to their cryo-EM determination by relating receptor activity to intramolecular interactions surrounding the embedded tethered ligand and lipid binding site. Finally, by describing two different G protein coupling modes for the same aGPCR, the authors widen our understanding of structural determinants driving receptor-mediated G protein signaling selectivity. In summary, this report allows for a unique cross-comparison of receptor/tethered ligand/G protein interface in an attempt to unify aGPCR structure-function relationship.

— We are grateful to the reviewer for the positive assessment.

While this reviewer finds this report to be nearly complete, a few minor details need to be addressed in a revised version of the manuscript:

1. Methods section: Please indicate the selected human variants of sequences used as templates for ADGRD1 and ADGRF1 mutants and WT.

— As suggested, the Uniprot numbers for the ADGRD1 and ADGRF1 variants have been added to the Methods section (lines 529-530, page 21).

2. Methods section: Please indicate which molecular biology method was used to generate receptor mutations.

— Site-directed mutagenesis PCR was used to generate the receptor mutations. This has been added to the Methods section as “All mutants used for structural and functional studies were generated by using site-directed mutagenesis PCR” (lines 542-543, page 21).

3. *Methods section: For BRET assays with stalk peptide, please indicate how the synthetic ligands were solubilized in order to be used in these assays (solvent, stock concentration, etc).*

— The details have been added to the Methods section as “The stalk peptides pD1 and pF1 were synthesized (GL Biochem, Shanghai), dissolved in DMSO at a concentration of 50 mM as stock solutions, and diluted to different concentrations with assay buffer (HBSS buffer (Thermo Fisher) supplemented with 20 mM HEPES, pH 7.4) upon assay” (lines 715-718, page 29).

4. *Methods section: Specify the exact biosensors per G protein family used in the methods and the corresponding conditions for transfection assays (which set of G protein subunits for each family of Gi and Gs were used, DNA amount used for each component including receptor plasmids, number of cells transfected per sample with biosensors and receptor DNA).*

— The details have been added to the Methods section as “The wild-type or mutant ADGRD1 and ADGRF1 were transiently co-transfected with plasmids encoding G α -RLuc8 (G α _{ss}-RLuc8 for G_s activation assay; G α _{i1}-RLuc8 for G_i activation assay), G β ₃ and G γ ₉-GFP2 at a ratio of 2:1:1:1 (receptor plasmid, 800 ng; G protein subunit plasmids, 400 ng for each) in 2 ml HEK293F cells at a density of 1.2×10^6 cells per ml” (lines 719-723, page 29).

5. *Methods section: Please indicate which DNA amount were used in order to obtain the constitutive activity described in cAMP and IP accumulation assays.*

— The amount of plasmids used in cAMP and IP accumulation assays has been added to the Methods section as “In brief, 2 ml HEK293F cells (Invitrogen; cells were routinely tested for mycoplasma contamination) at a density of 1.2×10^6 cells per ml were transiently transfected with 2,000 ng plasmid of the wild-type or mutant receptor and cultured at 37 °C for 48 h with 5% CO₂ atmosphere in a shaker shaking at 220 rpm” (lines 681-684, page 27).

6. *Methods section: It is unclear how the synthetic stalk peptides were obtained or generated. Please describe.*

— The stalk peptides were synthesized by a CRO company (GL Biochem, Shanghai). This has been clarified in the Methods (see point 3 above).

7. *Please provide a schematic of pF1 and pD1 -derived peptides used in the study.*

— A schematic of the pD1 and pF1 peptides has been added to Extended Data Fig. 1f.

8. *Fig 2 i,j: the figures indicates LS3 but should read LS6.*

— The typos have been corrected.

9. *Fig 2 i,j: For ease of reading, please reiterate/indicate the color legend attributed to either the stalk or receptor (non-stalk region).*

— The color legends are now shown in Fig. 2i, j and also clarified in the legend as “the bars are colored orange and green for the stalk and TMD mutations in ADGRD1, respectively (i), and colored magenta and blue for the stalk and TMD mutations in ADGRF1, respectively (j)”.

10. *Fig 2 i,j: please indicate stalk residues facing outward or inward beside the respective “Stalk-N” headers (Maybe “Stalk-N inward” and “Stalk-N outward”).*

— The suggestion is well taken. The stalk residues at different locations are further indicated with “Stalk-N inward” and “Stalk-N outward” in both the figure and legend of Fig. 2i, j.

11. *Fig 2g: Please include missing residue T810 in the ribbon structure.*

— The residue T810 has been added to Fig. 2g.

12. *Fig 2d, figure legend: please provide a short explanation of the mechanism for stalk-TMD interaction in absence of GAIN domain dissociation.*

— As suggested, a statement “The proteolysis is not required for stalk exposure that results in receptor activation and unfolding of the GAIN” has been added to the legend of Fig. 2d. More discussion of the mechanism for the proteolysis-independent activation is included in the main text (see response to Referee #2 below).

13. *Statistics: Given that the data in Fig 2i,j/Fig 3h,i/Fig 4c/were calculated from mean values, the variation should also be present in control samples used as a normalization reference. Please provide error bars corresponding to control samples.*

— The basal activity of the mutants from each independent experiment of IP and cAMP accumulation assays was calculated as per cent of the activity of the wild-type receptor measured in parallel. The basal activity of the control samples (wild-type receptors) was defined as 100% for each independent experiment, and thus has no error bar.

14. *Extended data Fig 5 and extended data table 3: It is unclear how the authors were able to determine EC50 values for curves which did not reach effect saturation. Please clarify in rebuttal and include the relevant information as part of extended data table 3 legend.*

— We thank the reviewer for this comment. Indeed, the concentration response curves of several mutants (ADGRF1-V755^{5.47b}A and R788^{6.37b}A; ADGRD1-L769^{6.49b}A, W773^{6.53b}A and F791^{7.42b}A) did not reach effect saturation within the concentration range tested (Extended Data Fig. 5). To avoid confusion, the data of these mutants in Extended Data Table 3 have been changed to “nd”, which is defined as “nd, not determined (data for which the concentration response curve could not reach effect saturation within the concentration range tested)” in the legend.

15. *Quantification of protein expression seems inaccurate in a few cases. For example, F641A mutant is reported at 19% of WT expression in extended data table 2 but at 82 % in extended data table 3, using the same flow cytometry protocol... Please revise these inconsistencies accordingly in the manuscript.*

— The protein expression levels in cell signaling (IP and cAMP accumulation, Extended Data Table 2) and G protein activation (BRET, Extended Data Table 3) assays were measured using the same flow cytometry protocol. However, the protein expression protocols were different for these two assays. Different amounts of receptor plasmids were used for transfection (IP/cAMP accumulation, 2,000 ng; BRET, 800 ng). In addition, co-expression with TRUPATH biosensors in the BRET assays may also influence the receptor expression.

Referee #2 (Remarks to the Author):

The manuscript by Qu and colleagues reports cryo-EM structures of two adhesion GPCRs, the ADGRD1 and the ADGRF1. While significant progress has been made over the past few years on structural understanding of GPCR function, the molecular basis of action of adhesion GPCRs remain poorly understood. The authors determined 4 cryo-EM structures of these two receptors to decipher how adhesion receptors are activated. A central question for the field has been how a tethered agonist, released by autoproteolysis of the GAIN domain, activates the 7TM domain of an adhesion receptor.

Especially confusing has been whether proteolysis is strictly required for adhesion GPCR activation, as several lines of evidence suggest that this is not needed per se. The major success of the authors work is that it reveals the active conformation of two adhesion receptors, in one case the same receptor bound to two engineered miniG proteins, thereby providing some insight into differential G protein coupling.

Perhaps the most interesting result is that a construct that cannot undergo proteolysis shows a tethered agonist conformation similar to the situation without the N-terminal fragment. The major weakness of the present manuscript is that it is written in a style where the figures and text largely report a laundry list of interactions, which is likely to be less interesting to a generalist audience. Regardless, the work is well done and thorough, with solid structural biology and functional studies.

— We are grateful to the reviewer for the positive assessment. As suggested, we have removed some details of stalk-TMD interactions mainly involved the stalk residues at positions of S2, S4 and S5 (paragraph 2, page 9), but included an extended discussion of the proteolysis-independent activation (see below).

While much of the mutagenesis proposed by the authors is useful, it largely confirms the interactions observed in the structures and provides little extra perspective for the field. I instead suggest that the authors revise the manuscript and put into context their very interesting observation that the uncleaved ADGRF1 is still in a similar conformation as the construct without the NTF. There is much to unpack here - is the expectation that activation leads to unfolding of the GAIN domain? Is there a stimulus required for this to happen? Do these constructs have a difference in constitutive activity; if so, how does one reconcile that with the structural observation?

— We thank the reviewer for the suggestion. As stated in the original version, we believe that the activation would lead to unfolding of the GAIN domain as the stalk is embedded within the core region of the GAIN and thus its dissociation, which is required for activation, most likely collapses the original folding of the GAIN. And as shown in Extended Data Table 2, the proteolysis-deficient mutants of both ADGRD1 and ADGRF1 have a wild-type (WT) level of basal activity in our functional assays (ADGRD1-H543A/T545A, 109 ± 8% of WT activity; ADGRF1-H565A/T567A, 99 ± 10% of WT activity). This data further supports that the proteolysis is not necessary for receptor activation. A stimulus that induces the autoproteolysis-independent activation may exist, such as a ligand that interacts with the extracellular domain(s) to facilitate the stalk exposure. However, more evidence is required for full understanding.

To provide extra perspective for the field and avoid over speculation, we have now included an extended discussion of the proteolysis-independent activation as “More intriguingly, a similar stalk-TMD interaction mode was also observed in the miniG_{i1}-bound structure of the proteolysis-deficient ADGRF1 (Extended Data Fig. 4h), demonstrating that the cleavage is not required for the stalk exposure and subsequent stalk-induced receptor activation (Fig. 2d). One possibility for this proteolysis-independent activation is that the receptor may exist at multiple conformational states, likely including a portion of receptor molecules with the stalk released from the GAIN domain, which leads to a collapse of the original folding of the GAIN. The dissociated stalk tends to interact with the receptor TMD to trigger G protein coupling, which in turn, stabilizes the stalk-TMD interaction on the extracellular side and may subsequently induce the stalk exposure of more receptor molecules by altering the equilibrium between different conformational states. An extracellular stimulus that facilitates the stalk exposure may exist, but more evidence is required for full understanding” (lines 152-163, page 7).